# Loss of consciousness reduces the stability of brain hubs and the heterogeneity of brain dynamics

Ane López-González [1,8✉], Rajanikant Panda[2,3,8], Adrián Ponce-Alvarez [1], Gorka Zamora-López [1], Anira Escrichs[1], Charlotte Martial[2,3], Aurore Thibaut[2,3], Olivia Gosseries [2,3], Morten L. Kringelbach [4,5,6], Jitka Annen [2,3], Steven Laureys [2,3,9] & Gustavo Deco [1,7,9]

Low-level states of consciousness are characterized by disruptions of brain activity that sustain arousal and awareness. Yet, how structural, dynamical, local and network brain properties interplay in the different levels of consciousness is unknown. Here, we study fMRI brain dynamics from patients that suffered brain injuries leading to a disorder of consciousness and from healthy subjects undergoing propofol-induced sedation. We show that pathological and pharmacological low-level states of consciousness display less recurrent, less connected and more segregated synchronization patterns than conscious state. We use whole-brain models built upon healthy and injured structural connectivity to interpret these dynamical effects. We found that low-level states of consciousness were associated with reduced network interactions, together with more homogeneous and more structurally constrained local dynamics. Notably, these changes lead the structural hub regions to lose their stability during low-level states of consciousness, thus attenuating the differences between hubs and non-hubs brain dynamics.

[1] Computational Neuroscience Group, Center for Brain and Cognition, Universitat Pompeu Fabra, Barcelona, Spain. [2] GIGA-Consciousness, Coma Science Group, University of Liège, Liège, Belgium. [3] Centre du Cerveau2, University Hospital of Liège, Liège, Belgium. [4] Department of Psychiatry, University of Oxford, Oxford, UK. [5] Center for Music in the Brain, Department of Clinical Medicine, Aarhus University, Aarhus C, Denmark. [6] Life and Health Sciences Research Institute, School of Medicine, University of Minho, Braga, Portugal. [7] Institució Catalana de la Recerca i Estudis Avançats (ICREA), Barcelona, Spain. [8] These authors contributed equally: Ane López-González, Rajanikant Panda. [9] These authors jointly supervised this work: Steven Laureys, Gustavo Deco. ✉email: ane.lopez@upf.edu

It is widely accepted that consciousness is decreased during sleep, under anaesthesia, or as a consequence of major brain lesions producing disorders of consciousness (DOC). In clinical settings, different states of consciousness have been defined depending on the level of wakefulness (i.e. arousal) and awareness (i.e. the content of consciousness)[1]. Wakefulness is usually evaluated by eye opening, and awareness by the responsiveness of the patients and their ability to interact with the environment, as a proxy for subjective experience. The study of these different levels of consciousness has proven to be essential to understand the neural correlates of consciousness, yet, the underlying mechanisms remain largely unknown. Elucidating these mechanisms is challenging since they seemingly rely on a non-trivial combination of alterations in local dynamics and network interactions.

During the last decades, the study of the organization of the brain and its dynamics has provided increased understanding of the healthy brain structure and function[2–7]. On the one hand, analyses of electroencephalography (EEG), functional MRI (fMRI), and magnetoencephalography (MEG) have shown that a hallmark of healthy awake brain dynamics is the balance between integration and segregation[8–11]. On the other hand, graph theory studies have shown that the modular and hierarchical organization of the human connectome facilitates the efficiency and robustness of information transmission[3,12]. For these reasons, consciousness has been considered to result from the interplay between dynamics and connectivity allowing the coordination of brain-wide activity to ensure the conscious functioning of the brain[13–16]. In contrast, unconscious states are characterized by a loss of integration[14,17,18], a loss of functional complexity[19,20], and a loss of communication at the whole-brain level[9,18,21,22]. Interestingly, it has been shown that functional connectivity deviates from structural connectivity during conscious wakefulness but it follows closer the organization of the anatomical connections during unconscious states[13,23–25]. Along with these global network effects, it has been proposed that some brain regions play an important role in maintaining consciousness, e.g. fronto-parietal regions, posterior cingulate, precuneus, thalamus and parahippocampus[1,26,27]. To study how structural, dynamical, local and network brain properties interplay in the different levels of consciousness, theoretical models are needed that incorporate all these levels of description.

In this study, we built whole-brain models with global and local parameters to investigate the possible mechanisms underlying the reduction of consciousness as a consequence of severe brain injury and transient physiological modifications due to propofol anaesthesia. For this, we studied the fMRI dynamics of patients who have suffered brain injuries from various etiologies (i.e. traumatic brain injury (TBI), anoxia, haemorrhage) affecting different brain regions implicated in DOC. Specifically, we analyzed data from patients with Unresponsiveness Wakefulness Syndrome (UWS; preserved arousal but no behavioural signs of awareness)[28] and in Minimally Conscious State (MCS; fluctuating but reproducible signs of consciousness)[29], and compared them with healthy control subjects (CNT) during conscious wakefulness i.e. resting-state. We also considered fMRI recordings of healthy controls scanned during conscious wakefulness (W), propofol-induced sedation (loss of responsiveness, S) and during the recovery from it (responsiveness regained, R). To study the brain dynamics, we performed phase-synchronization analyses that capture the relationships between the phases of BOLD signals at high temporal resolution — a method that has proven to effectively describe the spatiotemporal dynamics of fMRI signals[30].

To interpret these observations, we used a whole-brain model based on Hopf bifurcations[31]. This model combines single-node local oscillatory dynamics and network interactions. It is able to generate different collective dynamics depending on the shape of anatomical connectivity, the global strength of connections and the local state of the network's nodes. Importantly, the model allows investigating the interplay between the network structure and the dynamics at the local and global level. In particular, it allows us to study how network dynamics depend on the local activity of brain regions that have an important place in the structural network, such as highly connected nodes, usually referred to as "hubs".

## Results

We performed both data- and model-driven analyses to compare different levels of consciousness in two neuroimaging datasets comprising DOC patients and healthy subjects under propofol sedation. The first dataset consisted of fMRI signals and diffusion-MRI based structural connectivities (SC) from healthy subjects during conscious wakefulness ($n = 35$), and MCS and UWS patients ($n = 33$ and $n = 15$, respectively). The analysis was complemented with an additional fMRI dataset from 16 healthy controls scanned during conscious wakefulness (W), sedation (S) and recovery from it (R).

**Decreased brain phase dynamics complexity in low-level states of consciousness.** The first step in our analysis consisted of searching for spatiotemporal signatures of loss of consciousness from the whole-brain dynamics, as measured by the blood-oxygen-level-dependent (BOLD) signals. For this, we calculated the time-evolving functional connectivity based on the level of synchrony between the signals. The instantaneous phases of the BOLD were extracted in the 0.04–0.07 Hz frequency band[6,30,32] using the Hilbert transform (Fig. 1a, b). The phase-interaction matrix $P(t)$ was then defined as the pairwise phase differences between all regions of interest (ROIs, Fig. 1c). A $P(t)$ matrix is defined at every time point $t$, thus allowing us to define the phase-interaction matrices at the same temporal resolution as the BOLD. A variety of spatiotemporal properties were then quantified from the phase-interaction matrices.

We first examined the spatial properties of the phase-interaction matrices. To estimate the level of specialization and coordination in the network, we used measures of integration and segregation. The level of integration—cohesiveness in the network—was calculated by hierarchically scanning through the formation of connected components in the time-averaged phase-interaction matrix $\langle P \rangle$[8,9,33,34] (see "Methods"). To quantify segregation, we applied community detection methods on the matrix $\langle P \rangle$ to detect functional clusters of ROIs[35]. The quality of a partition of ROIs into clusters is evaluated by the modularity function (see "Methods"). A large modularity implies that ROIs are divided into well-defined clusters, indicating strong segregation. On one hand, we found that the average integration across time was significantly lower for MCS and UWS, compared to CNT, for S and R compared to W, and for S compared to R (Fig. 1d, see Table 1 for statistics). On the other hand, we found that the average segregation was significantly stronger for UWS compared to CNT, and for S compared to W (Fig. 1e, see Table 1 for statistics). Thus, low-level states of consciousness were characterized by a decrease of integration and an increase of segregation.

Second, we evaluated the temporal fluctuations of the mean phase-interaction. For this, at each time $t$, we computed the phase interaction averaged over ROIs, i.e., $r(t)$, see "Methods". The standard deviation of $r(t)$ provides an estimate of how much the average synchronization fluctuates in time. We found a significant reduction of phase-interaction fluctuations in low-level states of consciousness compared to conscious states (Fig. 1f;

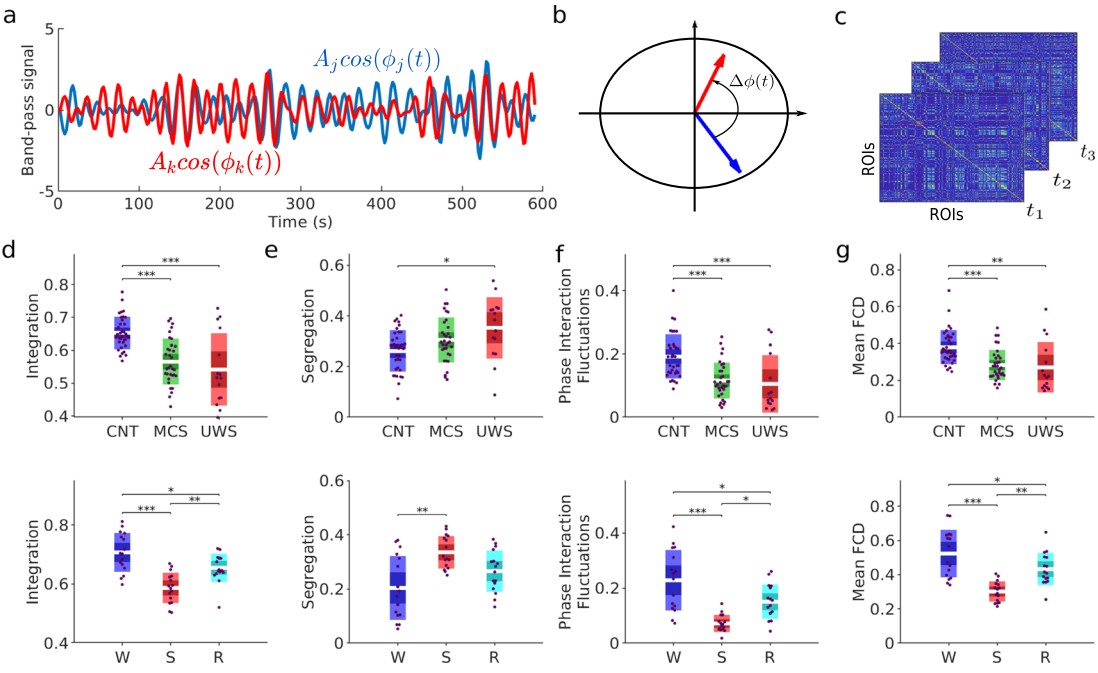

**Fig. 1 Changes in global properties of phase dynamics induced by loss of consciousness. a** BOLD band-pass signals (0.04–0.07 Hz) for two sample ROIs. The instantaneous phases, $\phi_j(t)$ and $\phi_k(t)$, of each signal were computed using the Hilbert transform. **b** At each time frame, the interaction between ROIs was given by the instantaneous phase difference, $\Delta\phi_{jk}(t) = |\phi_j(t) - \phi_k(t)|$, which can be represented as vectors in the unit circle of the complex plane. **c** Phase-interaction matrices $P_{jk}(t)$ were calculated as the cosine of the phase difference, $\cos(\Delta\phi_{jk}(t))$, at time $t$. All global measures used afterwards were based on the phase-interaction matrices. **d**–**e** The structure of phase interactions was described in terms of the integration and the segregation of the time-averaged phase interaction matrix (see "Methods"). **f** We quantified the temporal fluctuations of the mean phase synchrony (i.e., the average over ROIs of matrix $P(t)$) through its temporal standard deviation. **g** To detect the existence of recurrent synchronization patterns, we computed the FCD comparing phase-interaction matrices at different times (see "Methods"). Briefly, the FCD represents the (cosine) similarities between phase-interaction matrices at times $t$ and $t0$ for all possible pairs $(t, t0)$. The panel shows the average similarity for each experimental condition. In panels (**d**–**g**), each dot represents a participant and the boxes represent the measure's distribution. Boxplots represent the mean of the measures' values with a 95% confidence interval (dark) and 1 SD (light). Differences between groups were assessed using one-way ANOVA followed by FDR $p$-value correction. *: $p < 0.05$; **: $p < 0.01$; ***: $p < 0.001$ (see Table 1 for details).

**Table 1 Descriptive statistics and group comparisons of the global measurements of cerebral organization.**

|  | Integration | Segregation | Phase interaction fluctuations | Mean FCD |
|---|---|---|---|---|
| *DOC datatsets* |  |  |  |  |
| CNT | $0.65 \pm 0.01$ | $0.26 \pm 0.01$ | $0.15 \pm 0.01$ | $0.38 \pm 0.02$ |
| MCS | $0.56 \pm 0.01$ | $0.31 \pm 0.01$ | $0.09 \pm 0.01$ | $0.28 \pm 0.02$ |
| UWS | $0.54 \pm 0.03$ | $0.35 \pm 0.03$ | $0.07 \pm 0.02$ | $0.27 \pm 0.03$ |
| ANOVA | $p < 0.001$ | $p = 0.006$ | $p < 0.001$ | $p < 0.001$ |
|  | $F_{2,80} = 18.51$ | $F_{2,80} = 5.65$ | $F_{2,80} = 17.81$ | $F_{2,80} = 10.90$ |
| Multiple comparisons |  |  |  |  |
| $p_{CNT-MCS}$ | <0.001 | 0.207 | <0.001 | 0.014 |
| $p_{CNT-UWS}$ | <0.001 | 0.016 | <0.001 | 0.023 |
| $p_{MCS-UWS}$ | 0.241 | 0.223 | 0.420 | 0.594 |
| *Propofol anaesthesia datasets* |  |  |  |  |
| W | $0.71 \pm 0.02$ | $0.27 \pm 0.03$ | $0.14 \pm 0.03$ | $0.47 \pm 0.04$ |
| S | $0.59 \pm 0.01$ | $0.45 \pm 0.02$ | $0.07 \pm 0.01$ | $0.28 \pm 0.01$ |
| R | $0.65 \pm 0.01$ | $0.36 \pm 0.02$ | $0.12 \pm 0.02$ | $0.38 \pm 0.02$ |
| ANOVA | $p < 0.001$ | $p < 0.001$ | $p < 0.001$ | $p < 0.001$ |
|  | $F_{2,45} = 18.80$ | $F_{2,45} = 12.86$ | $F_{2,45} = 12.77$ | $F_{2,45} = 16.83$ |
| Multiple comparisons |  |  |  |  |
| $p_{W-S}$ | <0.001 | 0.001 | <0.001 | <0.001 |
| $p_{W-R}$ | 0.029 | 0.057 | 0.184 | 0.022 |
| $p_{S-R}$ | 0.006 | 0.039 | 0.014 | 0.017 |

The table shows the mean values and standard error of the empirical measures of integration, segregation, phase interaction fluctuations and mean FCD. Group comparison statistics were computed with a one-way ANOVA, followed by FDR correction (adjusted $p$-values are shown).

see also Table 1 for statistics), i.e. indicating that synchronization patterns in low-level states of consciousness fluctuate less than in conscious states.

Temporal fluctuations of the average phase-interaction matrix indicate excursions of the total level of synchrony over time but, alone, they do not capture the presence of connectivity patterns which re-occur over time. Therefore, we next evaluated the temporal recurrence of the phase-interaction matrices, referred as functional connectivity dynamics (FCD, see "Methods"), that describes how recurrent in time the synchronization patterns were. Briefly, this method computes the phase-interaction matrices averaged over sliding time windows and measures the similarity across all pairs of time windows, which is summarized in the FCD matrix. We found that low-level states of consciousness presented a significantly lower average FCD than in normal wakefulness (Fig. 1g; see also Table 1 and Supplementary Fig. S1). This suggests that the phase synchronization patterns were less recurrent in time for low-level states of consciousness.

Altogether, the above results show that, in both pathological and pharmacological low-level states of consciousness, the patterns of phase-synchronization were less connected, more segregated and less recurrent in time than in healthy conscious states.

**Decreased global coupling in low-level states of consciousness.** To gain insights into the possible mechanisms underlying the changes in the time-evolving functional connectivity reported in the previous sections, we used a computational model to describe the whole-brain dynamics. In this model, brain regions were modelled as oscillators, allowing the study of phase-synchronization patterns. Specifically, the local dynamics of individual brain regions were modelled by noisy Stuart-Landau oscillators, see Eqs. (7) and (8) in "Methods". This model captures the so-called Hopf bifurcation, a transition from noisy to oscillatory signals by the variation of a single parameter, and it has shown to fit the resting-state BOLD dynamics quite accurately[31]. In this case, the bifurcation parameter is the parameter $a_j$, representing the decay or growth rate of the system and thus controlling its stability. When $a_j < 0$, ROIs produce noisy signals and when $a_j > 0$ their signals become sustained oscillations, see Supplementary Fig. S2. At the transition, when $a_j \sim 0$, ROIs display flexible noisy oscillations of low amplitude, a regime in which ROIs are most susceptible to the inputs from other ROIs. The natural frequency of oscillations for each ROI was estimated from the peak of the power spectra estimated from their BOLD in the frequency band 0.04–0.07 Hz. Then, the $N = 214$ brain regions were coupled through the connectivity matrix $C_{jk}$, which is given by the structural connectivity of healthy subjects. The brain regions were defined according to the Shen atlas, ignoring the cerebellum[36]. The matrix $C_{jk}$ was scaled by a global coupling $g$. Thus, the large-scale network was weakly or strongly connected for small or large values of $g$, respectively (Fig. 2a). In summary, at this level of description the network dynamics depended on three ingredients: the local parameters for each node ($a_j$), the global strength of connections ($g$) and the network's structure ($C_{jk}$).

First, we studied the network dynamics for the homogeneous case, in which we set $a_j = 0$ for all nodes. This choice was based on previous studies which suggest that the best fit to the empirical data arises at the brink of the Hopf bifurcation where $a \sim 0$[31]. In this case, the network dynamics were determined by a single free parameter: the global coupling strength $g$. This parameter was estimated by fitting the FCDs from the empirical data with the FCDs calculated from the simulated signals at various values

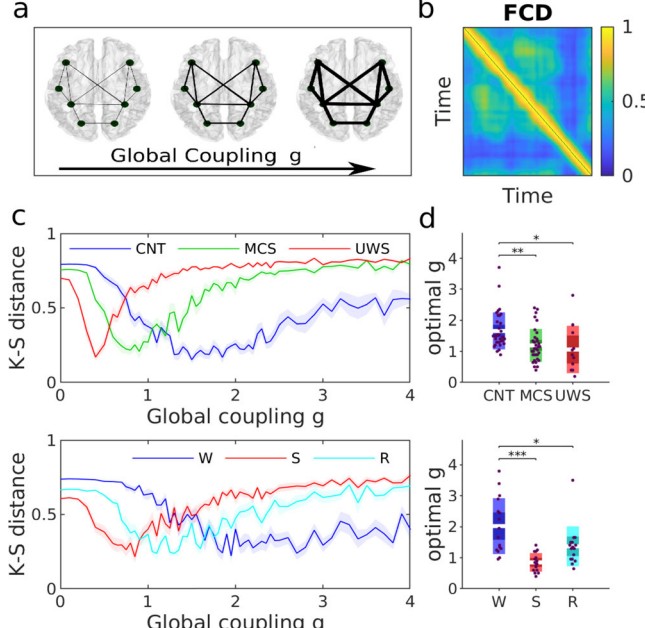

**Fig. 2 Fitting of global coupling parameter in the whole-brain network model. a** The global coupling model parameter $g$ scales the weights of the SC matrix. Low and high values of $g$ represent networks where the functional connectivity is weakly and strongly coupled to the structural networks, respectively. **b** To estimate this global parameter, we sought the model that best reproduced the distribution of FCD values (fixing all other model parameters). **c** KS-distance between the empirical and the model FCD distributions, as a function of $g$, for one participant of each subject group (top: healthy controls and DOC patients; bottom: awake and anaesthetized subjects). Solid lines and shaded areas represent the mean and the standard error of the fitting curves over simulation trials. **d** Optimal global coupling $g$ for all participants. In each panel, each dot represents a participant and the boxes represent the distribution of $g$. Boxplots represent the mean of the measures' values with a 95% confidence interval (dark) and 1 SD (light). Differences between groups were assessed using one-way ANOVA followed by FDR p-value correction. *: $p < 0.05$; **: $p < 0.01$; ***: $p < 0.001$. In panels (**c**) and (**d**), we used the healthy structural connectivity as the underlying connectivity of all models.

of $g$[31,37]. Specifically, empirical and simulated FCDs were compared using the Kolmogorov-Smirnov distance of their values (KS-distance, Fig. 2b). For low and high values of $g$, large KS distance indicates differences between the mean values of the FCD distributions. In the intermediate range of $g$ shorter KS distance evidenced a closer similarity between the empirical and the simulated FCDs (Fig. 2c). We considered the $g$ where KS distance is minimized as the optimal working point of the model[31,37,38]. Notably, although the fit of the model was based on the FCD, the models also maximized the fit of other data statistics including Pearson correlation functional connectivity and phase-interaction fluctuations (Supplementary Fig. S3).

We found that the optimal value of $g$ was smaller for states of low-level states of consciousness than for conscious wakefulness (Fig. 2c-d, see Table 2). This is consistent with the observation reported in the previous section that the correlation between structural and functional connectivity increases in states of low-level states of consciousness (Supplementary Fig. S4). The global coupling $g$ is a scaling parameter that controls for the conductivity of the fibres given by the SC. At low $g$ the network interactions are mainly restricted to ROIs directly connected by high strength links. Thus, increasing the global coupling favours the propagation of recurrent activity within the network allowing

**Table 2 Estimated global coupling parameters (means and standard deviations) for all experimental conditions.**

| Conditions | CNT | MCS | UWS | W | S | R |
|---|---|---|---|---|---|---|
| Global coupling $g$ | 1.7 ± 0.1 | 1.2 ± 0.1 | 0.8 ± 0.2 | 2.0 ± 0.2 | 0.9 ± 0.1 | 1.4 ± 0.2 |

p-values: $p_{CNT-MCS} = 0.015$, $p_{CNT-UWS} = 0.019$, $p_{MCS-UWS} = 0.7984$; $p_{W-S} < 0.001$, $p_{W-R} = 0.031$, $p_{R-S} = 0.080$. Values are explicitly provided for reproducibility, i.e. the simulations can be repeated with the global coupling strength's exact value.

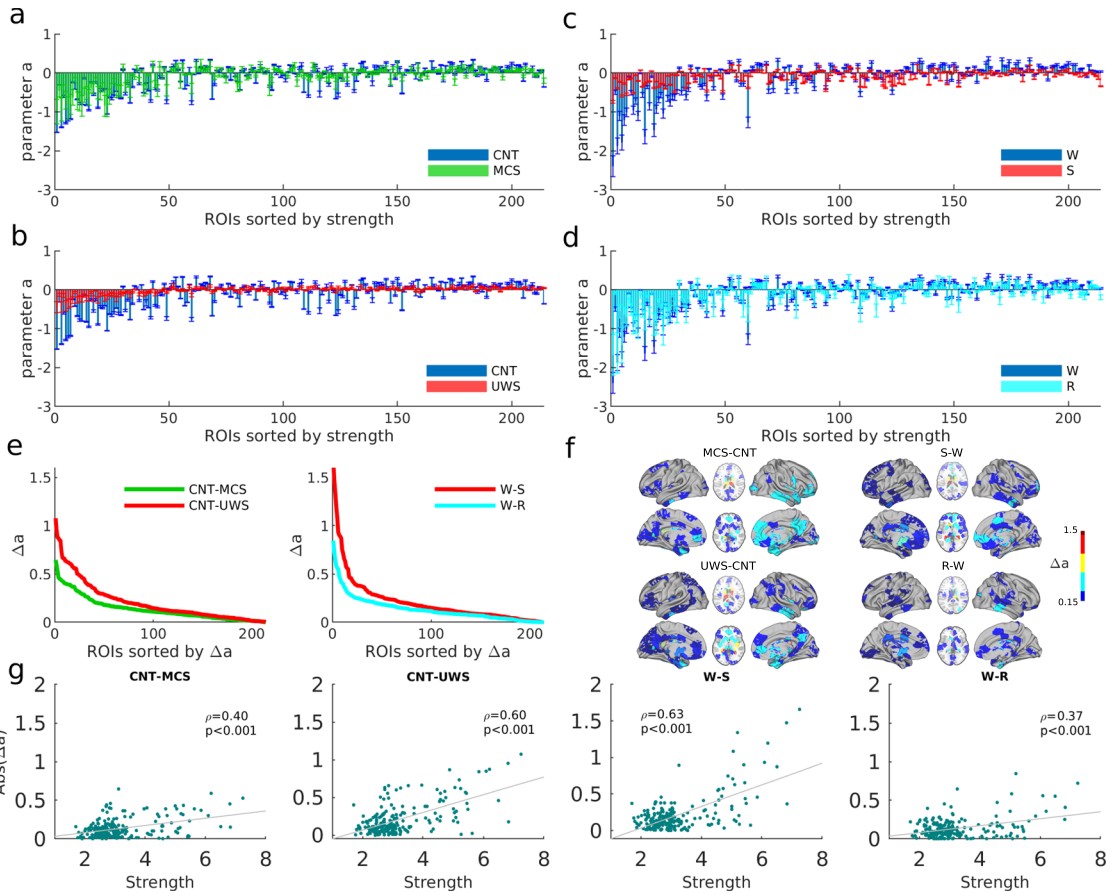

**Fig. 3 Local bifurcation parameters of the whole-brain model. a–d** Estimated bifurcation model parameters $a$ for each of the 214 nodes (sorted by node strength). Bars indicate the mean ± standard deviations across simulation trials. Results for low-level states of consciousness (MCS and UWS) are compared against the healthy controls in a and b. Results for anaesthesia and recovery (S and R states) are compared to the initial awake state (W) in c and d respectively. **e** Ranked absolute parameter difference, Δa, for all the comparisons. **f** Spatial distribution of Δa > 0.15 in the brain for each of the group comparisons. **g** Relationship between the absolute difference Δa and the strength of each node. The absolute difference of the parameter $a$ values between different groups as a function of the strength of the nodes extracted from the SC of the healthy controls. ρ corresponds to the Pearson correlation.

for correlations to emerge between nodes that are not directly connected with each other via structural connections. These results showed that in low-level states of consciousness, the brain dynamics were more constrained by the pairwise structural connections while in conscious awake —characterized by stronger levels of $g$— brain dynamics decouple from the purely direct anatomical constraints.

**Loss of regional heterogeneity in low-level states of consciousness.** We next asked whether we can obtain additional information by relaxing the homogeneity constraint on the local bifurcation parameters. In this case, the global coupling parameters $g$ were fixed to the ones estimated in the previous section—the homogeneous model in which all $a_j = 0$—but the local parameters $a_j$ were allowed to vary, thus introducing heterogeneity in the working point of the ROIs. The individual $a_j$ were estimated from the data using a gradient descent method (see "Methods").

We compared the resulting bifurcation parameters across nodes within and across groups. We found that bifurcation parameters in normal wakefulness (CNT and W groups) tended to be more negative as compared to those in low-level states of consciousness, Fig. 3a–d. This implies that the behaviour of ROIs in normal wakefulness are characterized by noisy oscillations— that are more stable—than their corresponding behaviour in low-level states of consciousness. This was especially the case for the dynamics of the structural hub ROIs, i.e. the nodes with the highest values of SC strength ($S_j = \sum_k C_{jk}$), which showed strong negative values of $a_j$ in normal wakefulness, Fig. 3a–d. Notice that in Fig. 3 ROIs are sorted by their SC strength in descending order. Comparing normal wakefulness (W) before and after sedation (recovery, R), both cases showed a similar distribution of $a_j$. In particular, the negativity of $a_j$ was reestablished for hubs in the recovery stage (Fig. 3d). This tendency was also observed when comparing MCS and UWS (Supplementary Fig. S5). We note that

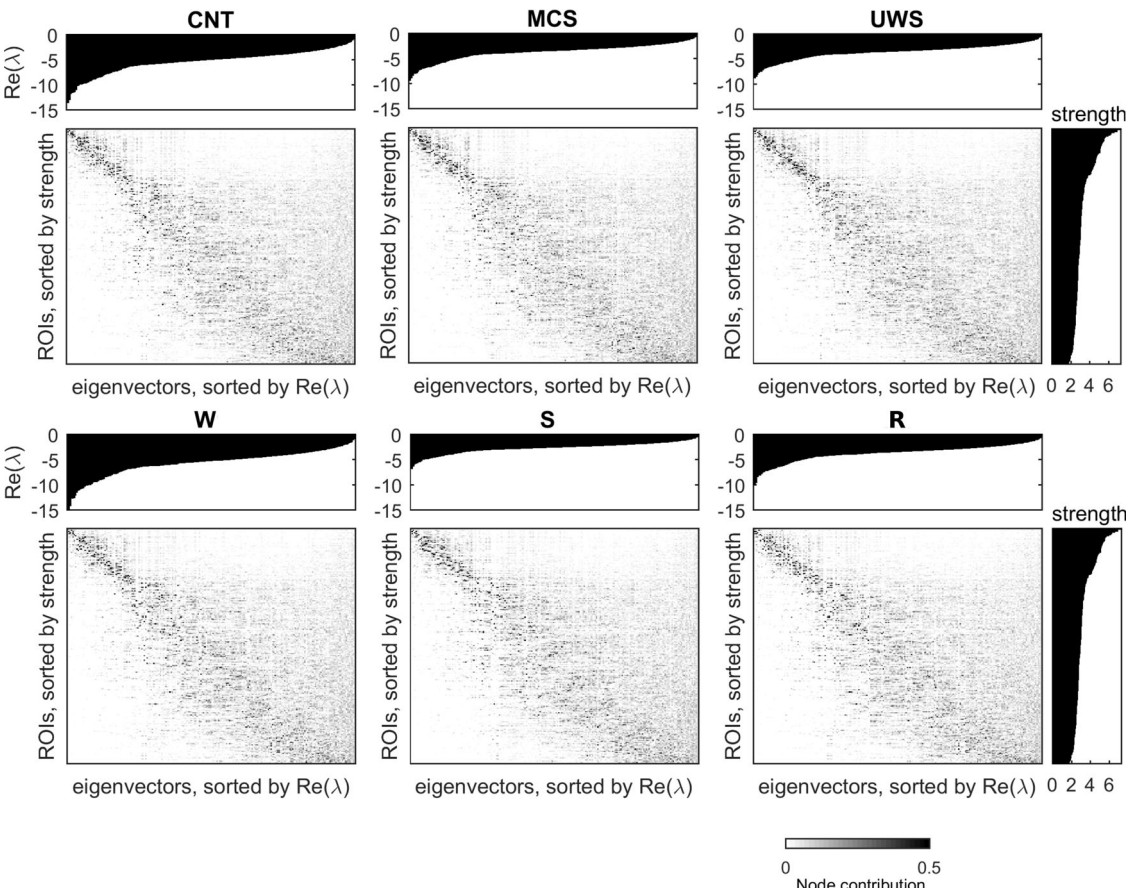

**Fig. 4 Eigendecomposition of the Jacobi matrix.** The eigenvectors of the Jacobi matrix ($N$-dimensional vectors) were sorted according to the real part of the associated eigenvalues (top insets), Real($\lambda$), and the strength of the nodes (right insets).

the resulting distribution of $a_j$ contributed to network collective dynamics, since shuffling the values of across brain regions lead to worse fits of the network statistics (Supplementary Fig. S6).

To identify the regions with the largest alterations in the local dynamical properties, we computed the absolute difference in local parameters, i.e. $\Delta a$, between patients and controls and between sedation/recovery and wakefulness (Fig. 3e–g). The largest differences in local parameters between controls and MCS/UWS patients were found in subcortical regions (thalamus, caudate, hippocampus and amygdala) and in some cortical regions (calcarine, insula, fusiform, frontal superior orbital, precuneus, cingulum, and temporal areas), see Fig. 3f left and Supplementary Tables S1-S2. When comparing the local parameters between wakefulness and sedation, the regions with largest differences included subcortical regions (thalamus, caudate, hippocampus, parahippocampal and putamen), and cortical regions (cingulum, insula, some frontal regions, paracentral and precentral), Fig. 3f top right and Supplementary Table S3. The main differences between wakefulness and recovery were found in the hippocampus, the cingulum and the precuneus, Fig. 3f bottom right and Supplementary Table S4. Interesting, $\Delta a$ and the connectivity strength of the brain regions significantly correlated (correlation: $0.40-0.94, p < 0.001$, Fig. 3g), indicating that the regions with the largest difference in the local bifurcation parameters were mostly the hubs.

Furthermore, we investigated the role of the regional dynamics in the stability of the system. For this, we studied the linear stability of the system by decomposing the Jacobi matrix **A** into eigenvectors (see "Methods"). Figure 4 shows the eigenvectors of **A** as a function of the node structural strengths and the real part

of the eigenvalues associated with the eigenvectors, Real($\lambda$). Since the system is stable, Real($\lambda$) < 0 for all eigenvalues. Clearly, the network hubs contribute the most to the most stable eigenvectors, i.e., those with lowest Real($\lambda$). This indicates that the hubs are key nodes to stabilize the system. Moreover, the stability of these dominant eigenvectors was reduced for models estimated from data corresponding to low-level states of consciousness, see Fig. 4. Thus, these results showed that the hubs lost their stabilizing role in low-level states of consciousness.

**Disentangling regional and network effects**. When observing the temporal activity of a brain region, as we do here via their BOLD signals, this activity is representative of the behaviour of the ROI embedded in the whole-brain network. In other words, we do not have access to the intrinsic activity of brain regions in isolation, as if they were separated from the rest of the network. Therefore, all ROI-specific parameters we estimated are necessarily affected by the network interactions. For example, the local bifurcation parameters presented in Fig. 3 incorporate effects coming both from the local dynamics and originated from the network interactions. In the following, we used a strategy to disentangle the changes in local parameters due to network effects from those due to local modifications. This analysis provides information about the origin (local or network-related) of the different dynamics of the ROIs for the different states of consciousness.

We defined an *effective* local parameter that is composed of the bifurcation parameter ($a_j$) and the connectivity strength of each node ($S_j = \sum_k C_{jk}$), given as $a_j^{\text{eff}} = a_j - gS_j$, (see "Methods"). For

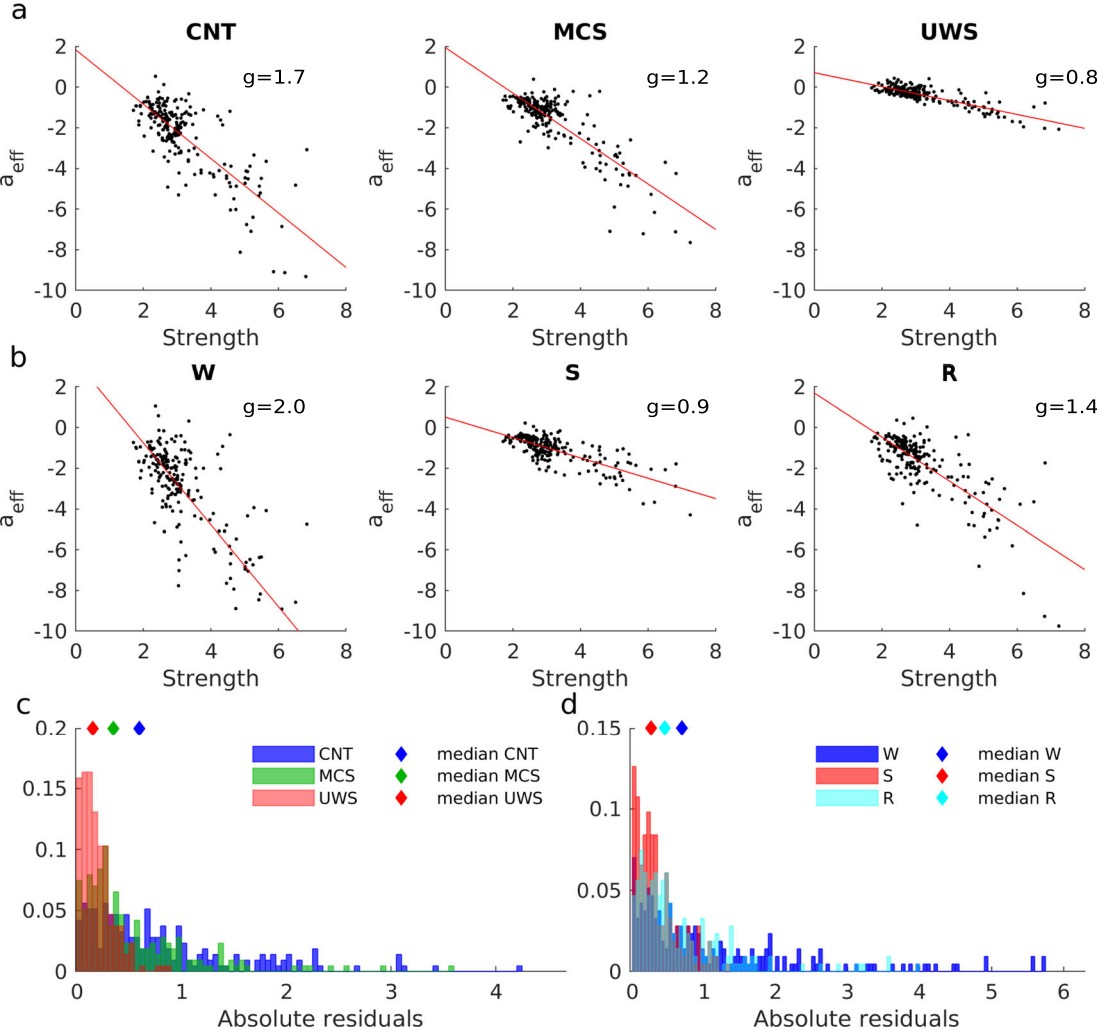

**Fig. 5 Disentangling structurally and dynamically driven heterogeneity of local nodes. a-b** The effective local bifurcation parameters, $a_j^{\text{eff}}$, were estimated using the heterogeneous model. In this model, the parameters $a_j^{\text{eff}}$ were optimized, after fixing $g$ to that obtained for the homogeneous model (see "Methods"). The obtained parameters were compared to the strengths of the nodes $S_j$, for healthy controls and DOC patients (**a**) and for awake and sedation conditions (**b**). In each panel, each dot represents one node. The red lines indicate the linear fits. **c** Distribution of the absolute residuals of each node given by the squared difference between the value of $a_j^{\text{eff}}$ and the estimated linear relationship between $a_j^{\text{eff}}$ and $S_j$, for each group. **d** Same as (**c**) but for W, S and R states.

the family of homogeneous models ($a_j$ = const.), the effective parameter is linearly related to the connectivity strength, while in the heterogeneous case deviations from this linear relation are to be expected. In other words, in the homogeneous case differences in effective local dynamics are fully explained by the network connections. In contrast, the heterogeneous model can produce additional diversity of local dynamics.

To disentangle the network effects from changes in local dynamics, we evaluated the deviations from the expected linear relation between effective local parameters and node strength in the different levels of consciousness. First, we estimated $a_j^{\text{eff}}$ from the data in each brain state using gradient descent with fixed $g$ for each condition (the values of $g$ were those of Fig. 2d). Note that, in this case, instead of estimating $a_j$, the method estimates directly $a_j^{\text{eff}}$ (see "Methods", Eq. 11). Next, we evaluated the linear regression deviation between $a_j^{\text{eff}}$ and the strength of the nodes (Fig. 5a, b). We found that the residuals of a linear regression were larger for control subjects and during healthy wakefulness than for DOC patients and sedation (Fig. 5c–d, $p < 0.002$ for all comparisons in both datasets computed with a one-way-

ANOVA, followed by FDR correction). These results indicated that conscious states are associated with collective dynamics emerging from regional heterogeneity—variance in the working point of the ROIs. In contrast, low-level states of consciousness are associated with dynamics generated by a network of ROIs with homogeneous working points. In this case, the observed dynamical differences across ROIs are predominantly explained by differences in connectivity strength. These results provided additional evidence that dynamics in low-level states of consciousness are strongly constrained by structural connections.

**Alteration of the structural core in DOC patients.** In the previous section, we have shown that under conditions of loss of consciousness the dynamical heterogeneity of the ROIs is reduced and that these changes affect specially the ROIs with largest node strength (Fig. 3a–d). In order to close the loop, our goal is now to investigate possible alterations in the structural connectivity of patients due to brain injuries that could cause the changes observed at the dynamic level. Therefore, we took a closer look at the hierarchical organization of the ROIs and their structural interconnections. We compare the strength of the ROIs—the sum

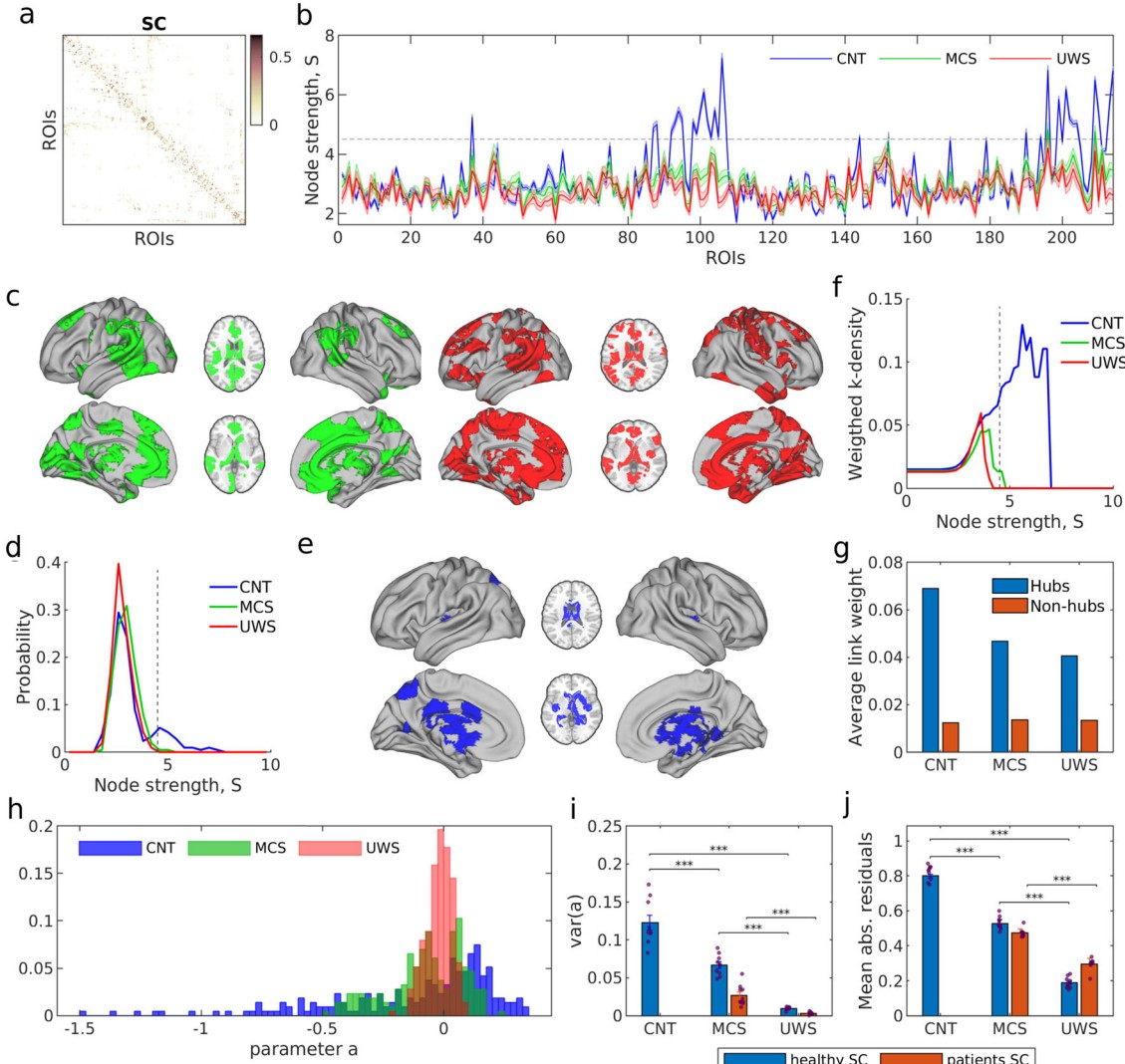

**Fig. 6 Disruption of the structural connectivity in DOC patients. a** SC matrices were averaged over subjects for each clinical group (CNT, MCS, and UWS). **b** Average node strength of each node for each group. Shaded areas represent the standard error across subjects. The grey dashed line corresponds to the threshold, $S = 4.5$, which determines the hub regions. **c** ROIs with significant decrease of strength in patients as compared to controls (Wilcoxon rank-sum test, followed by FDR correction). Left (green): CNT-MCS comparison; Right (red): CNT-UWS comparison. **d** Distribution of the node strength in the population average SCs. The distribution in controls displays a longer tail corresponding to hub regions, depicted in **e**. **f** k-density curves – average weight of links between regions with strength $S > S_0$ – show the loss of a rich-club structure in MCS and UWS patients. **g** Average link weight between hubs (i.e. regions with $S > 4.5$) in yellow and the average link weight between non-hubs regions (i.e. $S < 4.5$) in violet. **h** Distribution of the estimated bifurcation parameters $a_j$ using the average SC for each clinical group (healthy, MCS, and UWS). **i** The variance of the distribution of parameters $a_j$ for each clinical group. **j** Median of the absolute residuals of the linear relationship between the $a_{eff}$ vs strength. ***: $p < 0.001$, Wilcoxon rank-sum test, followed by FDR correction.

of connection weights per node—to search for highly connected ROIs (hubs). Notice that the models performed in the previous sections were constrained using the structural connectivity of the healthy subjects, see Fig. 6a. Using only the SCs of controls allowed comparing the optimal parameters of the model between groups and interpreting the alterations in the system.

The node strength across ROIs is heterogeneous in the control subjects with the strength of some ROIs notably standing out from the rest (Fig. 6b, blue line). However, the strength of ROIs in DOC patients (Fig. 6b, green and red lines) is rather homogeneous and no ROI stands out; implying that the presence of hubs is suppressed in DOC patients due to brain lesions. Regions with significantly decrease strength compared to controls ($p < 0.05$, Wilcoxon rank-sum test with FDR correction) in MCS

patients include the thalamus, the posterior and the anterior cingulum, hippocampus, the frontal medial, motor areas, caudate, precuneus, insula and precentral, for MCS patients (Fig. 6c left, see also Supplementary Table S5). In UWS patients, it included the aforementioned regions plus the fusiform, the parahippocampal, the cuneus, the lingual and the temporal areas (Fig. 6c right, and see details in Supplementary Table S6). Figure 6d shows the distribution of node strengths in the population average SCs for the three cases: control subjects (blue), MCS patients (green) and UWS patients (red). As seen, the presence of hub regions in the controls is characterized by a slow decay of the distribution at the higher end while the distributions for MCS and UWS patients rapidly decay. Hub regions with strength $S > 4.5$ in the controls are depicted in Fig. 6e. They comprise the insula,

thalamus, caudate, hippocampus, parahippocampal, calcarine, precuneus and cingulum mid.

Next, we examined the interconnections between the hub ROIs. A network is said to contain a rich-club if (i) it contains hubs and (ii) those hubs are densely interconnected among themselves forming a cluster. The presence of a rich-club is often regarded as the structural core that helps maintain the integration of cross-modular information, thus potentially facilitating consciousness[25,39,40]. As already shown, the presence of hubs is disrupted in the DOC patients and thus according to the first condition, without hubs no rich-club can be formed. We confirmed the disruption of the rich-club in DOC patients in two manners. First, we computed the weighted k-density on the population average SCs. This metric evaluates the average link weight between the regions with node strength $S$ larger than a given strength $S'$, scanning for all values of strength from zero to $S_{max}$. For healthy controls, the weighted k-density grown monotonically evidencing that the larger the strength of the nodes, the stronger were the links between them (Fig. 6f, blue curve). However, the lack of hubs in MCS and UWS patients led to an early cutoff of the k-density (green and red curves) evidencing the absence of a rich-club in these cases. Second, we compared the average link weight between the hub regions (previously selected from the SC of healthy controls as ROIs with $S > 4.5$) and non-hub regions, Fig. 6g. Clearly, the SC hubs were connected by stronger links than non-hubs in the three cases, and those links were strongest in the SCs of healthy controls.

Last, we remind that the whole-brain modelling performed in the previous sections were performed using the structural connectivity of the healthy subjects to constrain the model. We repeated the simulations for the cases of MCS and UWS patients using the injured connectomes from the patients. In these simulations, we did not find significant differences in the global coupling parameter $g$ in comparison with the values identified before (Supplementary Fig. S7). This was due to the high inter-individual variability of the structural connectivities. Also, consistent with the results above, we found that the heterogeneity of local bifurcation parameters was reduced for the models corresponding to DOC patients (Fig. 6h, i and Supplementary Fig. S8). Moreover, the dynamically based heterogeneity was significantly reduced as compared to the control case (Fig. 6j and Supplementary Fig. S9), indicating that local parameters were strongly determined by structural connections. These effects were stronger using injured SCs than using the healthy SC for all conditions, implying that structural damage in patients causes a rather homogeneous dynamical behaviour across ROIs.

## Discussion

In the present study, we have analyzed and modelled brain dynamics from patients with reduced consciousness due to brain damage (MCS and UWS), and from healthy participants under propofol-induced sedation. We have shown that reduction of consciousness is characterized by brain dynamics with less recurrent, less connected and more segregated patterns of phase-synchronization than for conscious states. Using whole-brain network models constrained upon healthy and injured structural connectivities, we could show that both pathological and pharmacological low-level states of consciousness present altered network interactions characterized by closer global resemblance between the phase synchronization patterns and the structural connectivity than in conscious wakefulness. Furthermore, low-level states of consciousness also manifest more homogeneous dynamical behaviour across regions. This effect was especially prominent in the structural hub regions—the most structurally connected ROIs—whose local dynamics shift towards unstable oscillatory regimes with a loss of their stability and disentangling

from the constraints to the anatomy in low-level states of consciousness.

The whole-brain network model used here allows us to understand how structural, dynamical, local and network properties interplay in the different levels of consciousness. Within this model, the network dynamics depend on three ingredients: (i) the global strength of connections, (ii) the regional bifurcation parameters and (iii) the organization of the structural connectivity. We allowed these features to vary in the different model versions we used here. First, assuming homogeneous local dynamics across brain regions, we found that low-level states of consciousness had lower global coupling strength than conscious states. This finding is consistent with the observation that functional connectivity decreases in states of low-level states of consciousness[13,24,39] and it explains why in this case functional connectivity follows closer to structural connectivity. Indeed, the global coupling is a scaling parameter that controls for the conductivity of the structural connections in the model. At low coupling, the propagation of activity is mainly restricted to ROIs connected by links with large strength. Increasing the global coupling favours the propagation of activity through direct and indirect connections within the network, thus allowing for correlations to emerge also between nodes that are not directly connected with each other.

Second, we used a model where the local bifurcation parameters ($a$) were allowed to vary individually for each region. These parameters were estimated from the data, resulting in a more heterogeneous distribution of their values in conscious wakefulness than in low-level states of consciousness. In particular, we found that during conscious wakefulness the behaviour of structural hubs is characterized by noisy oscillations ($a < 0$) that are more stable than for the rest of the regions. In contrast, in low-level states of consciousness, all regions display oscillations close to the transition ($a \sim 0$) without differentiation between hubs and non-hubs. Interestingly, linear stability analysis showed that the stable noisy oscillations of the hubs primarily determine the network stability. These results suggest that in order to release the structural constraints on local dynamics, while ensuring the global stability of the system, hubs play an important role by diminishing their variability. In contrast, unstable hubs would propagate noise to the rest of the network, thus degrading the communication among brain regions. Furthermore, we showed that differences in local parameters could arise by different local dynamics or by different connectivity to the rest of the network. We disentangled these two possible origins of variability by estimating the effective local parameters. Our analysis showed that, for low-level states of consciousness, the estimated local dynamics were strongly determined by the structural connections, impeding any additional heterogeneity arising from dynamics, which is consistent with the weaker coupling previously discussed.

The relevance of the results presented here shall be framed under various aspects. Theories of consciousness such as Integrated Information Theory[11] or the Global-Workspace Theory[10,26] propose that higher-level associations and consciousness require the dynamic integration of sensory information processed previously by specialized brain regions (segregation). Thus, an imbalance of this coexistence between integration and segregation could lead to different pathologies. Consistent with this view, we found an alteration of integration-segregation of functional phase interactions during low-level states of consciousness caused by brain damage, propofol anaesthesia, and anaesthesia's long-lasting effects during recovery (Fig. 1). Here, we showed that the diversity of phase synchronization patterns and their recurrence in time were also reduced in low-level states of consciousness, presumably leading to a failure

to dynamically balance integration and segregation. These results are in line with previous studies showing differences in the synchronized states both in space and time during altered states of consciousness[14,17,18,24,39–42].

From an anatomical point of view, the study of brain connectivity in the recent decades has shown that large-scale structural connectivity is modular and hierarchically organized, with the multiple communication pathways centralized by a set of highly connected brain regions (the hubs) that are densely interconnected forming a rich-club[7,43,44]. This architecture, also known as core-periphery networks, is expected to facilitate the coexistence of integration and segregation of information in the brain. It has been proposed that the imbalance between integration and segregation can lead to loss of consciousness[17,45], impairing the neural communication across specialized brain modules or subnetworks[46–48] or impeding the integration of that information by the core hubs. Here, we showed that structural breakdown of core-periphery architecture, as observed in injured structural connectivity, leads to a reduction of dynamical heterogeneity (Fig. 5). Including the damaged structural connectivities due to brain injuries in the DOC patients into the model showed a further limitation of the diversity of local dynamics in pathological low-level states of consciousness.

From a dynamical point of view, our results show that the breakdown of an additional dynamic-based heterogeneity observed in conscious states leads to an attenuation of the stability properties of the hubs during low-level states of consciousness. We believe that the dynamical stability of the hubs is a signature of consciousness and has functional implications. Indeed, the stability of hubs is required to maintain a functional core-periphery architecture. Such core-periphery architecture is essential to achieve a trade-off between stability and flexibility[49], with the network periphery supporting more responsivity and plasticity while the network core aids in maintaining the robustness of the system[50–53]. Consistent with this view, previous works on whole-brain fMRI have observed core-periphery organization during resting state[54] and a stable core together with a variable periphery during learning[55]. In conclusion, we find that functional disruption in low-level states of consciousness might partly be caused by an attenuation of core-periphery structure induced by (i) the structural damage of the hubs or (ii) the loss of stability of the hubs.

Overall, our results suggest that, during healthy wakefulness, in order to allow a dynamically based heterogeneity of local dynamics across the brain, resulting in diverse collective activity patterns, while preserving stability and a core-periphery architecture, the hubs are required to "anchor" the dynamics by increasing their stability.

It is well known in clinical literature that loss or reduction of consciousness is related to the impairment of certain key brain regions that are dynamically and/or structurally altered during low-level states of consciousness. These areas have been proposed previously to be involved in the thalamo-cortical loop and are thought to down-regulate the activity of the cortical network which is impaired in loss of consciousness[1,26,27,56]. The results presented here evidence that the regions characterized by altered dynamical and structural properties coincide with the structural hubs. Among these areas, we found stronger effects in subcortical areas, such as the thalamus and hippocampus, and the precuneus and the posterior cingulate areas, for both pathological and physiological low-level states of consciousness. Our results not only indicate the areas affected by the loss or reduction of consciousness but also give a mechanistic explanation, such as loss of heterogeneity, loss of stability and higher constraint to the anatomy, of the underlying brain dynamics in low-level states of consciousness.

Indeed, the methodology and results presented here provide new insights into the understanding of the brain network behaviour after applications of interventions. The observed decrease in global connectivity is consistent with previous studies of EEG signals after a transcranial magnetic stimulation (TMS)-mediated perturbation, showing that the brain was less responsive in low-level states of consciousness than in conscious states[19,20,57]. A prediction of our study is thus that, under localized external stimulation, hub regions should be less responsive during conscious states compared to low-level states of consciousness. A current hypothesis in the field is that the enhancement of neural excitability in the affected regions through therapeutic procedures may improve the conscious recovery process[58]. However, current stimulation protocols using TMS to investigate the network response during different states of consciousness in humans[19,20,57] cannot achieve the required localization of the perturbation propagation to test our predictions. TMS is a strong external perturbation that indirectly activates several cortical and subcortical areas, producing a global perturbation of ongoing activity. Furthermore, the measurement of the response using EEG is not described with enough spatial resolution to measure the effect on hubs directly. Nevertheless, at the moment, in-silico perturbation of diverse computational models[33,59] might be useful to test this prediction.

Our study is restricted to the comparison between conscious wakefulness and low-level states of consciousness, i.e. DOC patients and propofol-induced anaesthesia state, which are distinguished by the levels of awareness and wakefulness. Interestingly, although the underlying physiology for loss of consciousness differs between DOC patients and propofol sedation, the dynamics at the whole-brain level and the alterations in local dynamics seem to be similar. Future work should study in more detail the differences in the local mechanism altering the global state of consciousness. Indeed, the relatively similar phenomenology[60] of the two different states may have a shared cellular basis, at the level of pyramidal neurons, underlying the observed alterations in the global dynamics[61,62].

Given the patient inclusion criteria of the present study, generalization of our results to a broader spectrum of DOC patients, such as those presenting larger brain structural damage, remains to be corroborated. Future studies should consider the confirmation of the results to other anaesthetics agents besides propofol, such as ketamine and sevoflurane, whose effect takes place through different molecular pathways. Also, other theories have proposed a multi-dimensional definition of consciousness which include additional factors such as visual perception, cognition or/and experience of unity[63]. Those dimensions show different levels in states of altered consciousness, such as under psychedelic drugs or meditation. In the light of our results, we would expect that under a drug-induced psychedelic state, where the conscious content seems to increase and the brain shows higher entropy in the local firing rates[64], the whole-brain models will show an increase of heterogeneity and a larger decoupling from the structural connectivity, while the hubs should lose the stability present during resting state and the entropy associated with the repertoire of states would increase.

In this study, we intended to study a whole-brain model that is able to produce oscillations, as needed to represent the synchronization statistics of the data. Thus, we chose the Stuart–Landau model to characterize the local, regional dynamics. This model represents the normal form of a Hopf bifurcation, i.e., the universal behaviour around a bifurcation producing oscillation through a limit-cycle. Despite its simplicity and non-biological origin, the model has shown to generate a rather accurate fit to the BOLD dynamics, beyond the success of other models in the past[31,37,65]. However, many alternative mean-field

or population models exist that could be chosen for the regional dynamics. Therefore, the generalization of the results here presented shall be confirmed in future studies, which employ different local models.

Using global synchronization measures, we found significant differences for different levels of consciousness (CNT and DOC patients and W, S, and R), but these measures mostly failed to identify a significant difference between patients groups (MCS vs. UWS) (Fig. 1). However, our model-based analysis of local dynamics was able to distinguish between patients groups (Figs. 3e–g, 5c, d, 6i, j). This highlights the clinical translation potential of multi-parameter whole-brain models and the need for further studies that consider region-specific measures for clinical predictions.

Electrophysiological, fMRI and MEG studies have shown that heterogeneous local dynamics, differing between sensory and association brain regions, contribute to the hierarchical specialization across areas at the functional level[66–70]. Recently, it has been shown that extending models to include heterogeneous information of local dynamics, e.g., as given by positron-emission tomography (PET) maps of neurotransmitter receptor density[68] or by $Tw1/Tw2$ maps as proxies of microcircuit properties[69], increases model performance to fit empirical data. Our model could be extended to include these and other axes of hierarchy to explore the brain mechanism of consciousness.

In conclusion, our results show that pathological and pharmacological low-level states of consciousness presented altered network interactions, more homogeneous, structurally constrained local dynamics, and less stability of the network's core compared to conscious states. These results provide relevant information about the mechanisms of consciousness both from the theoretical and clinical point of view.

## Methods
**Participants**. This study includes a cohort of healthy controls and patients suffering from disorder of consciousness (DOC), and a cohort of healthy subjects undergoing anaesthesia-induced loss of consciousness. The study was approved by the Ethics Committee of the Faculty of Medicine of the University of Liège. Written informed consent to participate in the study was obtained directly from healthy control participants and the legal surrogates of the patients.

We selected 48 DOC patients, 33 in MCS (9 females, age range 24–83 years; mean age ± SD, 45 ± 16 years) and 15 with UWS (6 females, age range 20-74 years; mean age ± SD, 47 ± 16 years) and 35 age and gender-matched healthy controls (14 females, age range 19-72 years; mean age ± SD, 40 ± 14 years). The DOC patients data was recorded 880 ± 35 days after injury. The healthy controls data was collected while awake and aware. The diagnosis of the DOC patients was confirmed through repeated behavioural assessment with the Coma Recovery Scale-Revised (CRS-R) that evaluates auditory, visual, motor, sensorimotor function, communication and arousal[71]. The DOC patients were included in the study, if MRI exam was recorded without anesthetized condition and the behavioural diagnosis was carried out at least five times for each patient using CRS-R examination[72]. 5 CRS-R assessments were performed within a period of 14 days, usually within one week. The best CRS-R diagnosis was used for clinical diagnosis. One CRS-R assessment was performed before the MRI acquisition on the same day, yet the clinical diagnosis was made based on the best out of 5 CRS-R's. The exclusion criteria of patients were as follows: (i) having any significant neurological, neurosurgical or psychiatric disorders prior to the brain insult that lead to DOC, (ii) having any contraindication to MRI such as electronic implanted devices, external ventricular drain, and (iii) being not medically stable or large focal brain damage, i.e. >2/3 of one hemisphere. Details on patients' demographics and clinical characteristics are summarized in Supplementary Table S7-S8.

For the propofol anaesthesia, 16 healthy control subjects (14 females, age range, 18–31 years; mean age ± SD, 22 ± 3.3 years) were selected in three clinical states including normal wakefulness with eyes closed (W), propofol anaesthesia-induced sedation (S) and recovery from propofol anaesthesia (R). Propofol was infused through an intravenous catheter placed into a vein of the right hand or forearm and an arterial catheter was placed into the left radial artery. During the study ECG, blood pressure, SpO2 and breathing parameters were monitored continuously. Sedation was achieved using a computer-controlled intravenous infusion of propofol to obtain constant effect-size concentrations (for details on the procedure, see ref. [73]). The propofol plasma and effect-size concentrations were estimated using the three-compartment pharmacokinetic model[74]. After reaching the appropriate effect-site concentration, a 5-min equilibration period was allowed

to insure equilibration of propofol repartition between compartments. Arterial blood samples were then taken immediately before and after the scan in each clinical state for subsequent determination of the concentration of propofol and for blood-gas analysis. The level of consciousness was evaluated clinically throughout the study with the Ramsay scale[75]. The subject was asked to strongly squeeze the hand of the investigator; she/he was considered fully awake or to have recovered consciousness if the response to verbal command ('squeeze my hand!') was clear and strong (Ramsay 2), in mild sedation if the response to verbal command was clear but slow (Ramsay 3), and in sedation if there was no response to verbal command (Ramsay 5–6). For each consciousness level assessment, Ramsay scale verbal commands were repeated twice. Before and after each scanning session, a reaction time task was also performed to provide additional information on the clinical state of the volunteers. Three clinical states were defined in this study: normal wakefulness (Ramsay 2), sedation (Ramsay 5) and recovery of consciousness (Ramsay 2). It should be noted that during the recovery of consciousness, R, subjects showed clinical recovery of consciousness (i.e., same score on Ramsay sedation scale as during wakefulness) but they showed residual plasma propofol levels and lower reaction times scores. The healthy subjects did not have MRI contradication, any history of neurological or psychiatric disorders or drug consumption, which have significant effects in brain function. It shall be noted that the anaesthesia dataset has a gender imbalance of 70–30% female to male.

**MRI acquisition and data analysis**. For the healthy controls and DOC patients, structural and functional MRI (fMRI) data were acquired on a Siemens 3T Trio scanner (Siemens Inc, Munich, Germany). The BOLD fMRI resting state (i.e. task free) was acquired using EPI, gradient echo with following parameters: volumes = 300, TR = 2000 ms, TE = 30 ms, flip angle = 78°, voxel size = 3 × 3 × 3 mm³, FOV = 192 × 192 mm², 32 transversal slices, with a duration of 10 minutes. Subsequently, structural 3D T1-weighted MP-RAGE images with were acquired with following parameters: 120 transversal slices, TR = 2300 ms, voxel size = 1.0 × 1.0 × 1.2 mm³, flip angle = 9°, FOV = 256 × 256 mm². Last, diffusion weighted MRI (DWI) was acquired in 64 directions (b-value =1,000 s/mm², voxel size = 1.8 × 1.8 × 3.3 mm³, FOV = 230 × 230 mm², TR = 5,700 ms, TE = 87 ms, 45 transverse slices, 128 × 128 voxel matrix) preceded by a single unweighted image (b0). The DWI was acquired twice.

The propofol dataset was acquired on a 3T Siemens Allegra scanner (Siemens AG, Munich, Germany). The fMRI resting state were acquired using the following parameters: EPI, gradient echo, volumes = 200; TR = 2460 ms, TE = 40 ms, voxel size = 3.45 × 3.45 × 3 mm³, FOV = 220 × 220 mm, 32 transverse slices, 64 × 64 × 32 matrix size. The structural images were acquired using 3D T1-weighted MP-RAGE with following parameters: 120 transversal slices, TR = 2250 ms, TE = 2.99ms, voxel size = 1 mm³, flip angle = 9°, FOV = 256 × 240 × 160mm.

Preprocessing of MRI data was performed using MELODIC (Multivariate Exploratory Linear Optimized Decomposition into Independent Components) version 3.14[76], which is part of the FMRIB's Software Library (FSL, http://fsl.fmrib.ox.ac.uk/fsl). Preprocessing steps included: discarding the first 5 volumes, motion correction using MCFLIRT[77], non-brain removal using BET (Brain Extraction Tool)[78], spatial smoothing with 5 mm FWHM Gaussian Kernel, rigid-body registration, high pass filter cutoff = 100.0 s, and single-session ICA with automatic dimensionality estimation. After preprocessing, FIX (FMRIB's ICA-based X-noiseifier)[79] was applied to remove the noise components and the lesion-driven artefacts, independently, for each subject, see Supplementary Fig. S10. Specifically, FSLeyes package in Melodic mode was used to manually classify the single-subject Independent Components (ICs) into "good" for signal, "bad" for noise or lesion-driven artefacts and "unknown" for ambiguous components. Each component was classified by looking at the spatial map, the time series, and the temporal power spectrum[80,81]. Finally, FIX was applied by using the default parameters to obtain a cleaned version of the functional data.

FSL tools were used to obtain the blood-oxygen-level-dependent (BOLD) time series of the 214 cortical and subcortical brain regions (without the cerebellum, see more details in Supplementary Table S9) in each individual's native EPI space, defined according to a resting-state atlas[36]. Specifically, the cleaned functional data previously obtained were co-registered to the T1-weighted structural image by using FLIRT[82]. Then, the T1-weighted image was co-registered to the standard MNI space by using FLIRT (12 DOF) and FNIRT[82,83]. The resulting transformations were concatenated and inverted and applied to warp the resting-state atlas from MNI space to the cleaned functional data. To ensure the preservation of the labels, a nearest-neighbour interpolation method was used. Then, the BOLD time series for each of the 214 brain regions were extracted for each subject in their native space by using fslmaths to obtain a binary mask of each brain region, and fslmeans to obtain the time series of each binary mask.

The grand average of the functional connectivity matrix, FC, was constructed using Matlab 2017 (The MathWorks Inc.) to compute the pairwise Pearson correlation between all 214 brain regions, applying Fisher's transform to the r-values to get the z-values for the final 214 × 214 functional connectivity matrices.

**Structural connectivity**. A whole-brain structural connectivity (SC) matrix was computed for each subject from the DOC dataset, using two-step process as described in previous studies[84–86]. Similar to the procedure used for analyzing the

resting-state fMRI data, we used the resting-state atlas to create a structural connectivity in each individual's diffusion native space. First, DICOM images were converted to Neuroimaging Informatics Technology Initiative (NIfTI) format using dcm2nii (www.nitrc.org/projects/dcm2nii). The b0 image in DTI native space was co-registered to the T1-weighted structural image by using FLIRT[82]. The T1-weighted structural image was co-register to the standard space by using FLIRT and FNIRT[82,83]. The resulting transformations were inverted and applied to warp the resting-state atlas from MNI space to the native MRI diffusion space by applying a nearest-neighbour interpolation algorithm. Second, analysis of diffusion images was performed using the processing pipeline of the FMRIB's Diffusion Toolbox (FDT) in FMRIB's Software Library (www.fmrib.ox.ac.uk/fsl). The non-brain tissues were extracted by applying the Brain Extraction Tool (BET)[78], the eddy current distortions and head motion were corrected using eddy correct tool[87], and the gradient matrix was reoriented to correct for subject motion[88]. Then, Crossing Fibres were modelled using the default BEDPOSTX parameters and the probability of multi-fibre orientations were calculated to improve the sensitivity of non-dominant fibre populations[89,90]. Then, Probabilistic Tractography was performed in native MRI diffusion space using the default settings of PROBTRACKX[89,90]. For each brain region, the connectivity probability to each of the other 213 brain regions was computed. The resulting matrix was then symmetrized by computing their transpose matrix and averaging both matrices, $C_{jk} = C_{kj}$. Finally, to obtain the structural probability matrix, the value of the probability pairs of brain regions was divided by its corresponding number of generated tracts. To summarize, for each participant, a $214 \times 214$ symmetric weighted network was constructed and normalized by the total number of fibres in the whole network; thus, the structural connectivity matrix (SC) represents the density of links of the anatomical organization of the brain.

**Phase-interaction matrices.** To evaluate the level of synchrony in brain activity, the phase interaction between BOLD signals was evaluated. Therefore, a band-pass filter within the narrowband of $0.04-0.07$ Hz was applied in order to extract the instantaneous phases $\phi_j(t)$ for each region $j$. This frequency band captures more relevant information than other frequency bands in terms of brain function[32]. The instantaneous phases, $\phi_j(t)$, were then estimated applying the Hilbert transform to the filtered BOLD signals individually. The Hilbert transform derives the analytic representation of a real-valued signal given by the BOLD time series[31]. The analytical signal, $s(t)$, represents the narrowband BOLD signal in the time domain. This analytical signal can be also described as a rotating vector with an instantaneous phase, $\phi(t)$, and an instantaneous amplitude, $A(t)$, such that $s(t) = A(t) \cos(\phi(t))$. The phase and the amplitude are given by the argument and the modulus, respectively, of the complex signal $z(t) = s(t) + i \cdot H[s(t)]$, where $i$ is the imaginary unit and $H[s(t)]$ is the Hilbert transform of $s(t)$.

The synchronization between pairs of brain regions was characterized as the difference between their instantaneous phases. At each time point, the phase difference $P_{jk}(t)$ between two regions $j$ and $k$ was calculated as

$$P_{jk}(t) = \cos\left(|\phi_j(t) - \phi_k(t)|\right). \tag{1}$$

Here, $P_{jk} = 1$ when the two regions are in phase ($\phi_j = \phi_k$), $P_{jk} = 0$ when they are orthogonal and $P_{jk} = -1$ when they are in anti-phase. At any time $t$, the phase-interaction matrix $P(t)$ represents the instantaneous phase synchrony among the different ROIs. The time-averaged phase-interaction matrix, $\langle P \rangle = \sum_{t=1}^{T} P(t)/T$, was bias-corrected by subtracting the expected phase-interactions phase-randomized surrogates, designed to decorrelate the phases while preserving the power spectrum of the original signals (see Surrogate Analysis section).

The instantaneous global level of synchrony of the whole network $r(t)$ was calculated as the average of the phase differences at each time point. Since $P(t)$ is a symmetric matrix, then:

$$r(t) = \frac{1}{N(N-1)} \sum_{j=1}^{N} \sum_{k=j+1}^{N} P_{jk}(t). \tag{2}$$

Finally, the fluctuations of $r(t)$ over time indicate the diversity of the observed network phase interactions. The *phase-interaction fluctuations, m,* were thus calculated as the standard deviation of $r$. When all the nodes of a network are synchronized then $r(t) = 1$ for all $t$ and thus $m = 0$. However, if the network switches among synchronization states over time leading to fluctuations of $r$, then $m > 0$, reflecting those fluctuations.

**Surrogate analysis.** For the phase surrogate analysis, first, the Fourier transform (FT) of the signals was computed. The phase of the Fourier transform was substituted with uniformly distributed random numbers while preserving their modulus. Then, the inverse FT was applied to return to the time domain with the new Fourier coefficients. This procedure effectively randomizes the phases of the signals while preserving the same power spectra as the original time-courses. Specifically, let $x_j(t)$ be the original BOLD time-course from the brain area $j$. The discrete Fourier transform of $x_j$ is given by:

$$\tilde{x}_j(k) = \sum_{t=1}^{T} x_j(t) e^{-i\frac{2\pi kt}{T}} \tag{3}$$

where $i$ is the imaginary unit and k goes from 1 to T ($k = 1, ..., T$). The phase

shuffled surrogate is given by:

$$x_j^{\text{surr}}(t) = \frac{1}{T} \sum_{k=1}^{T} |\tilde{x}_j(k)| e^{-i\left(\frac{2\pi kt}{T} + \varphi_r\right)} \tag{4}$$

where $\varphi_r$ is random variable uniformly distributed between $-\pi$ and $\pi$. These surrogates were used to rerun the analysis and extract a phase interaction matrix used to correct the empirical matrices.

**Integration.** Integration refers to the capacity of the brain to maintain communication between different parts and subnetworks. Here, we employed a metric of integration that assesses the connectivity out of the phase-interaction matrix, scanning across different scales[8,9,33,34]. More precisely, the time-averaged phase-interaction matrix, $\langle P \rangle$, is scanned through all possible thresholds ranging from 0 to 1. At each threshold, the matrix is binarised and the size of its largest connected component is identified. Integration is then estimated as the integral of the size of the largest connected component as a function of the threshold.

**Segregation.** Segregation refers to the breakdown of a system into functional subcomponents. Quantitatively, segregation was estimated by the modularity index $Q^{91}$ of the time-averaged functional connectivity matrix $\langle P \rangle$, which is the metric evaluating how good a community detection method in networks could separate the network into modules. Therefore, we first binarized the matrix $\langle P \rangle$ by detecting the pairs of regions with average phase interaction significantly ($p < 0.01$) larger than expected in phase-randomized surrogates. Second, the Louvain algorithm was employed to subdivide this matrix into modules. The Newman modularity Q of the optimal partition was then considered as the measure of segregation[35]. Modularity is a cost function that evaluates the quality of subdivisions of networks into modules by targeting the maximization of the number edges within the modules and thus the minimization of edges across them[35]. Thus, the modularity index on the functional connectivity is a reasonable representation of the subdivision of the brain's activity into functional subdivisions.

**Functional connectivity dynamics (FCD).** We evaluated the presence of repeating patterns of network states by calculating the recurrence of the phase-interaction patterns. For this, we used the functional connectivity dynamics (FCD). This measure is based on previous studies that defined the FCD for FC matrices calculated in different time windows[92]. In our study, the duration of scans (10 min) was divided into sliding windows of 30 time points, shifted in 2 s steps. For each time window, centred at time $t$, the average phase-interaction matrix, $\langle P(t) \rangle$, was calculated as

$$\langle P(t) \rangle = \frac{1}{T} \sum_{|t-t'|<15} P(t'), \tag{5}$$

where $T$ is the total number of TRs. We then constructed the $M \times M$ symmetric matrix whose $(t_1, t_2)$ entry was defined by the cosine similarity, $S_{\cos}$, between the upper diagonal elements of two matrices $\langle P \rangle(t_1)$ and $\langle P \rangle(t_2)$, given as

$$S_{\cos}(t_1, t_2) = \frac{\vec{p}(t_1) \cdot \vec{p}(t_2)}{|\vec{p}(t_1)||\vec{p}(t_2)|} = \cos(\theta), \tag{6}$$

where $\vec{p}(t_1)$ and $\vec{p}(t_2)$ are the vectorized representations of matrices $\langle P(t_1) \rangle$ and $\langle P(t_2) \rangle$, respectively, and $\theta$ corresponds to the angle formed between the two vectors, $\vec{p}(t_1)$ and $\vec{p}(t_2)$. Finally, the FCD measures were given by the distribution of these cosine similarities for all pairs of time windows.

**Whole-brain network model.** The brain network model consists of $N = 214$ coupled brain regions derived from the Shen parcellation[36]. The global dynamics of the brain network model used here results from the mutual interactions of local node dynamics coupled through the underlying empirical anatomical structural connectivity matrix $C_{jk}$[31]. Local dynamics are simulated by the normal form of a supercritical Hopf bifurcation, i.e., Stuart–Landau oscillator[93,94], describing the transition from noisy oscillations to sustained oscillations[95], and is given, in the complex plane, as

$$\frac{d\mathbf{z}}{dt} = (\mathbf{a} + i\omega) \odot \mathbf{z} - (\mathbf{z} \odot \bar{\mathbf{z}})\mathbf{z} + \beta\mu(t), \tag{7}$$

where $\odot$ is the Hadamard element-wise product, $\mathbf{z} = [z_1, \ldots, z_N]$ are the complex-valued state variables of each node, $\bar{z}$ is the complex conjugate of $z$, $\mathbf{a} = [a_1, \ldots, a_N]$ and $\boldsymbol{\omega} = [\omega_1, \ldots, \omega_N]$ are the vectors containing the bifurcation parameters and intrinsic frequencies of each node in the range of 0.04–0.07 Hz band, respectively, and $\boldsymbol{\mu} = [\mu_1, \ldots, \mu_N]$ is a Gaussian noise vector with standard deviation $\beta = 0.02$ based on previous studies[31,37,65]. The intrinsic frequencies were estimated from the averaged peak frequency of the narrowband empirical BOLD signals of each brain region. For $a_j < 0$, the local dynamics present a stable spiral point, producing damped or noisy oscillations in the absence or presence of noise, respectively (Supplementary Fig. S2). For $a_j > 0$, the spiral becomes unstable and a stable limit-cycle oscillation appears, producing autonomous oscillations with frequency $2\pi f_j = w_j$. The BOLD fluctuations were modelled by the real part of the state variables, i.e., Real($z_j$).

The whole-brain dynamics were obtained by coupling the local dynamics through the $C_{jk}$ matrix:

$$\frac{dz_j}{dt} = z_j[(a_j + i\omega_j) - |z_j|^2] + g\sum_{k=1}^{N} C_{jk}(z_k - z_j) + \beta\mu_j(t), \quad (8)$$

where $g$ represents a global coupling scaling the structural connectivity $C_{jk}$. The matrix $C_{jk}$ is scaled to a maximum value of 0.2 to prevent the full synchronization of the model. Interactions were modelled using the common difference coupling, which approximates the simplest (linear) part of a general coupling function[96].

*Homogeneous model: Fitting the global coupling g.* To create a representative model of BOLD activity in each brain state, we adjusted the model parameters ($g$ and $a_j$) to fit the spatiotemporal BOLD dynamics for each brain state and each dataset. Our first aim was to describe the global properties of the spatiotemporal dynamics of each subject in each state, independently of the variations in the dynamics of local nodes. For that reason, in this first approach to the model, all nodes were set to $a_j = 0$, called the homogeneous model. The global coupling parameter $g$ was obtained by fitting the simulated and empirical data. Specifically, for each value of $g$, the model FCD was computed and compared with the empirical FCD using the Kolmogorov-Smirnov (KS) distance between the simulated and empirical distribution of the FCD elements. The KS-distance quantifies the maximal difference between the cumulative distribution functions of the two samples. Thus, the optimal value of $g$ was the one that minimized the KS distance.

*Heterogeneous model: local optimization of the bifurcation parameters.* To evaluate the heterogeneous local dynamics on the network's dynamics, we extended the model to allow differences in bifurcation parameters $a_j$ for different ROIs. The $g$ parameter was the one estimated with the homogeneous model. The bifurcation parameters were optimized based on the empirical power spectral density of the BOLD signals in each node. Specifically, we fitted the proportion of power in the 0.04–0.07 Hz band with respect to the 0.04–0.25 Hz band (i.e. we removed the smallest frequencies below 0.04 Hz and considered the whole spectrum up to the Nyquist frequency which is 0.25Hz)[31,37]. For this, the BOLD signals were filtered in the 0.04–0.25 Hz band and the power spectrum $PS_j(f)$ was calculated for each node $j$. We then defined the proportion of power in the 0.04–0.07 Hz band as

$$p_j = \frac{\int_{0.04}^{0.07} PS_j(f)df}{\int_{0.04}^{0.25} PS_j(f)df} \quad (9)$$

We updated the local bifurcation parameters by an iterative gradient descendent strategy, i.e.:

$$a_j^{new} = a_j^{old} + \eta(p_j^{emp} - p_j^{sim}), \quad (10)$$

until convergence. $\eta$ was set to 0.1 and the updates of the $a_j$ values were done in each optimization step in parallel.

*Relation between the strength of a node and its dynamics.* Finally, the relation between local and network dynamics was studied. An effective bifurcation parameter $a_j^{eff}$ was defined which contains information of the local dynamics and local structure given by its strength. This parameter permits to extract the relation between the dynamics and structure of each node. Note that the effective bifurcation parameter refers to the perturbed local dynamics including network effects (not to be confused with *effective* connectivity). More specifically, in Eq. (8), we separated the part that relates to the effective local dynamics and the part that relates to the interaction between nodes. Noting that $\sum_{k=1}^{N} C_{jk}(z_k - z_j) = \sum_{k=1}^{N} C_{jk}z_k - z_j\sum_{k=1}^{N} C_{jk}$, Eq. (8) can be written as

$$\frac{dz_j}{dt} = \left(a_j - g\sum_{k=1}^{N} C_{jk} + i\omega_j\right)z_j - z_j|z_j|^2 + g\sum_{k=1}^{N} C_{jk}z_k + \beta\mu_j(t). \quad (11)$$

Taking $a_j^{eff} = a_j - g\sum_{k=1}^{N} C_{jk}$, we obtain

$$\frac{dz_j}{dt} = \left(a_j^{eff} + i\omega_j\right)z_j - z_j|z_j|^2 + g\sum_{k=1}^{N} C_{jk}z_k + \beta\mu_j(t). \quad (12)$$

Note that, if $a_j$ is homogeneous across the network ($a_j = a$ for all $j$), $a_j^{eff}$ is linearly related to the nodal strength $S_j = \sum_{k=1}^{N} C_{jk}$.

**Linear stability analysis.** In this section we studied the linear stability of the whole-brain network. The model consists of 214 coupled brain regions, with local Hopf dynamics, coupled through the connectivity matrix **C**. The dynamical system presented in Eq. (8) and considering the coupling to the structural connectivity can be written in vector form as

$$\frac{d\mathbf{z}}{dt} = (\mathbf{a} - g\mathbf{S} + i\omega) \odot \mathbf{z} - (\mathbf{z} \odot \bar{\mathbf{z}})\mathbf{z} + g\mathbf{Cz} + \beta\mu(t), \quad (13)$$

where $\mathbf{S} = [S_1, \dots, S_N]$ is the vector containing the strength of each node, i.e. $S_j = \sum_{k=1}^{N} C_{jk}$. The model parameters **a** and **ω** were estimated from the data, using the heterogeneous model, and $g$, using the homogeneous model, for each experimental condition.

We studied the linear stability of the fixed point $\mathbf{z} = 0$, which is solution of $\frac{d\mathbf{z}}{dt} = 0$. In the linearized system the quadratic terms (i.e., $\mathbf{z} \odot \bar{\mathbf{z}}$) are not taken into account and the evolution of fluctuations $\delta\mathbf{z}$ around $\mathbf{z} = 0$ can be approximated as

$$\frac{d}{dt}\delta\mathbf{z} = \mathbf{A}\delta\mathbf{z} + \beta\mu(t), \quad (14)$$

where **A** is the Jacobi matrix, given as $\mathbf{A} = \text{diag}(\mathbf{a} - g\mathbf{S} + i\omega) + g\mathbf{C}$, and $\text{diag}(\mathbf{x})$ is the diagonal matrix whose entries are the elements of the vector **x**.

**Graph analysis of the structural connectivity.** The network organization of the SC matrices was investigated using measures of graph theory (GAlib: Graph Analysis library in Python/Numpy, www.github.com/gorkazl/pyGAlib). We focused only on the potential presence of hub regions and a rich-club to relate these structural features to the observed dynamical properties of the brain regions. Given a connectivity matrix **C** with entries $C_{jk}$ indicating the weight of the link between nodes $j$ and $k$, the strength of a node ($S_j$) is defined as the sum of the connections it makes: $S_j = \sum_{1}^{N} C_jk$. A rich club is a supra-structure of a network happening when (i) a network contains hubs and (ii) those hubs are densely interconnected with each other, forming a cluster[97]. Identifying the presence of a rich-club typically implies the evaluation of k-density, $\rho(k)$, an iterative process which evaluates the density $\rho(k')$ of the remaining part of network after all nodes with degree $k<k'$ have been removed[97]. Here, we employed the version of the metric adapted for weighted networks, iterating from node strength $S' = 0$ to $S' = S_{max} = 10$ in steps of $\Delta S = 0.2$. At each iteration step, the average link weight $\rho(S')$ between the nodes with strength $S>S'$ was computed.

**Statistical analysis.** Statistical differences between levels of consciousness were assessed using one-way repeated measures (rm) ANOVA followed by multiple comparisons using False Discovery Rate (FDR) correction[98]. The threshold for statistical significance was set to $p$-values < 0.05. Wilcoxon rank-sum test (equivalent to a Mann–Whitney U test) was applied in order to find region-wise differences between CNT and DOC patients in the strength of the SC. We corrected for multiple comparisons by using the FDR correction, considering $P < 0.05$ as statistically significant.

**Reporting summary.** Further information on research design is available in the Nature Research Reporting Summary linked to this article.

## Data availability
The data can be requested to the Authors and examples of the phase-interaction matrices of functional connectivity of different states of pharmacological and pathological states of consciousness are available on the Knowledge Graph (Human Brain Project) https://search.kg.ebrains.eu/instances/Dataset/775c7858-2305-4a56-8bd6-865c4ab5dd4f.

## Code availability
The MATLAB code of the phase-synchronization measures and whole-brain models are available on Github (https://github.com/decolab/Hopf_consciousness) and the python codes of the structural connectivity analysis are available on (GAlib: Graph Analysis library in Python/Numpy, https://github.com/gorkazl/pyGAlib).

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

## Acknowledgements
A.L.G. and G.D. were supported by Swiss National Science Foundation Sinergia grant no. 170873. A.P.A. and G.D. received funding from the FLAG-ERA JTC (PCI2018-092891). G.D., S.L. and G.Z.L. received funding from the European Union's Horizon 2020 Framework Programme for Research and Innovation under the Specific Grant Agreement No. 785907 (Human Brain Project SGA2) and No. 945539 (Human Brain Project SGA3). G.D. acknowledges the Spanish Ministry Project PSI2016-75688-P (AEI/FEDER), the Catalan Research Group Support 2017 SGR 1545, and AWAKENING (PID2019-105772GB-I00, AEI FEDER EU) funded by the Spanish Ministry of Science, Innovation and Universities (MCIU), State Research Agency (AEI) and European Regional Development Funds (FEDER). The study was further supported by the University and University Hospital of Liège, the Belgian National Funds for Scientific Research (FRS-FNRS), the European Space Agency (ESA) and the Belgian Federal Science Policy Office (BELSPO) in the framework of the PRODEX Programme, "Fondazione Europea di Ricerca Biomedica", the Bial Foundation, the Mind Science Foundation and the European Commission, the fund Generet, the King Baudouin Foundation, AstraZeneca foundation, and the DOCMA project [EU-H2020-MSCA-RISE-778234]. R.P. is research fellow, O.G. is research associate and S.L. is research director at F.R.S.-FNRS. M.L.K. is supported by the ERC Consolidator Grant: CAREGIVING (no. 615539), Center for Music in the Brain, funded by the Danish National Research Foundation (DNRF117), and Centre for Eudaimonia and Human Flourishing funded by the Pettit and Carlsberg Foundations.

We would like to thank the healthy participants and the patients, their families, caregivers and treating clinicians for their participation in this study. The authors thank the whole staff from the ICU and Nuclear Medicine departments, University Hospital of Liège. We are highly grateful to the members of the Liège Coma Science Group for their assistance in clinical evaluations.

## Author contributions
A.L.G., A.P.A., G.Z.L., S.L. and G.D. designed research. R.P., J.A., A.T., C.M. and O.G. acquired the data. A.L.G., R.P., A.E. and M.K. preprocessed the data. A.L.G. and R.P. analyzed the data. A.L.G., A.P.A., G.Z.L., M.K. and G.D. studied the computational model. A.P.A., G.Z.L., O.G., S.L. and G.D. supervised research. A.L.G., R.P., A.P.A., A.E. and G.Z.L. wrote the manuscript. All authors contributed to the editing of the manuscript.

## Competing interests
The authors declare no competing interests.
