## [Peer Review File · Communications Biology]

Reviewers' comments:

Reviewer #1 (Remarks to the Author):

1 Manuscript overview

The authors present a study of changes in global properties of phase-dynamics in fMRI data sets and model-based from patients that suffered brain injuries leading to a disorder of consciousness and from subjects undergoing propofol-induced anaesthesia. Their study is focused on studying how structural, dynamical, local and network brain properties interplay in the different levels of consciousness. The authors claim that pathological and pharmacological low-level states of consciousness displayed less recurrent, less diverse, less connected, and more segregated synchronization patterns than conscious states.

The authors study these effects using whole-brain models built on healthy and injured connectomes, showing that altered dynamics arise from a global reduction of network interactions, together with more homogeneous and more structurally constrained local dynamics. These effects were accentuated using injured connectomes.

In my opinion, the work is interesting, well written, and presents enough scientific merit.

However, the current state of the manuscript presents different aspects that can improve. In my opinion, the work needs to address major and minor issues before further consideration.

2 Major

- In my opinion the title of the article does not summarize the main findings. The hubs are not well emphasized in the text as the definition only comes in the legend of figure 5. Additionally, it is not clear to me that as you used it is a genuine topological feature. On the other hand, the stability analysis only appears at the end of the supplementary material.
- Again in the same line. The article refers 28 times to the supplementary material and important results including one on the title (stability) is in the supplementary material. I believe that either the title should be changed or the article should emphasize results in the main text. Otherwise should balance and reorganize the main text to include what you believe is the most important finding. Figures one and three summarize what you are really doing in this article that is to study using fMRI data sets and model-based data changes in local and global properties of phase-dynamics induced by loss of consciousness.
- At the end of the introduction the authors refer to topological aspects, however the definition of Hubs as nodes with high strengths is not topological (derived from an adjacency matrix). While later in the text seems that methodologically you end up binarizing the SC still seems to me that you are not referring to a topological property.
- There is a mismatch in the definition of strength in line 201 and 536 and what you consider in the linear stability analysis in section 1.4. You consider in the definition of strength the parameter G or not?
- There exist several alternative whole-brain models to study the brain. The Jansen & Rit model and Dynamic Mean-Field are among them. Alternative modeling approaches can be used to study Integration, Segregation, and Mean FCD. I'm not asking for further analysis but the article should explain more clearly why you are focusing on Stuart Landau oscillators rather than more biophysically grounded alternatives.
- There exist alternative definitions of Integration and Segregation. This fact should be emphasized and the choice better motivated.
- I can understand that some sensitive data cannot be shared. However, in my opinion, the code used to obtain the results should always be available, readable, and usable without even need to request it.

3 Minor

- In formula 4 there is a \cos missing, and I will recommend to use $k \cdot k$ instead of $| \cdot |$
- The formula of angle ... dot product or internal product.
- The authors used a unidimensional notion of consciousness, i.e. states can be mapped from low-levels to high-levels of consciousness, but there are some other perspectives that argue that consciousness is better characterized as a multidimensional construct (see <https://www.ncbi.nlm.nih.gov/pmc/articles/PMC7167214/>). Even when the results shows that different states can be mapped onto a single dimension, a small discussion on how their results relate (or not) to a multi-dimensional account for consciousness could highly enrich the article.
- The selection of colors in the boxplots is not appropriate to see the values of all the subjects,

specially in the red and the blue boxes.

- Formula 1 in the supplementary material is t going from $1..T$?
- Figure S6 should consider a line and an inset with the p values in each subplot.
- In my estimation you can do better using the information of the tables S1 and S2, probably a good idea would be to compare the ROIs with highest absolute difference in bifurcation parameter between DOC and anaesthesia as it seems to have an interesting overlap.
- Considering that you are counting the number of nodes within the largest component I would refer to the size of the largest component not the length.
- Line 134. The name of the atlas could be mentioned in the results for clarity.
- I only know graphs, subgraphs and hypergraphs. What is a super-graph?
- line 365 Liège.
- line 424 $z(t) = s(t) + i \cdot H[s(t)]$
- There is a typo on line 471.
- Discussion, line 317. The authors argue that the reduced variability and increased stability of hubs are essential for consciousness. However, modeling and empirical results show that in the psychedelic state many brain hubs increase their entropy (i.e. increased variability, see for example <https://www.nature.com/articles/s41598-020-74060-6>). During the psychedelic state consciousness is not lost, rather, conscious content seems to increase, so, how can these results on psychedelic states could be integrated in the framework presented by the authors? Is the psychedelic state an example where the stability of hubs is reduced, but consciousness is preserved ?
- Discussion. Line 354. Any ideas on how anesthesia and DOC states converge onto similar dynamics despite their divergences on the underlying physiological mechanisms? A small discussion on the possible physiological mechanisms that make these two unconscious states similar in terms of the departure from conscious wakefulness dynamics would greatly improve the reach of the article. See for example [https://www.cell.com/trends/cognitive-sciences/fulltext/S1364-6613\(20\)30175-3](https://www.cell.com/trends/cognitive-sciences/fulltext/S1364-6613(20)30175-3).
- In the supp material section 1.4 you refer to Figure S6 when should refer to Figure S8

Reviewer #2 (Remarks to the Author):

The authors investigated fMRI brain dynamics from patients showing disorder of consciousness following brain injuries and from subjects showing altered consciousness due to anaesthesia. The altered states of consciousness observed were different from not altered states of consciousness. Starting from these observations they declare that developed models (homogenous and heterogenous) to investigate the mechanisms (in terms of global and local dynamics) underlying the loss of consciousness. The authors interpreted their results in a whole-brain model framework constrained by a structural connectome. Despite the idea is by itself interesting, some methodologies are vaguely presented, contain inaccuracies and an appropriate critical discussion in terms of other works with same techniques is missing. This is especially true for the graph theory and complex networks analysis tools. In what follows are the details. The topic that the authors declare to address is of general interest in the field of the biology. However, the main goal of the paper is not convincingly reached in my opinion.

I have some general comments regarding the paper and some suggested revisions:

General comments:

In general, I think that the paper is too dense and rich of analysis that are not well explained, and gives the impression that the main statements lack of scientific clarity. Results presentation is not accurate and self-consistent, and some logical leaps are present. The frequent references both to the Methods section and to the supplementary material make difficult to follow the author's claims precisely and generate confusion. Furthermore, the continuous references to many of the supplementary figures leave the doubt that many of them are be crucial to properly understand the results. A non-expert reader would not be able to disentangle the bundle of information reported. This is also the case of the Figures: the captions are often confused and not well planned in terms of content. These aspects require the reader to jump continuously affecting his ability to focus on the main messages. Furthermore, I found out that some of the important concepts (and measure and indexes) are not clearly defined (f.e. the concept of Hubs that appears in the title),

and or the definition of the employment of a certain measure is not always justified. Some sections of the Methods are imprecise and hasty, especially those regarding graph theory tools.

I have a main concern regarding the selection of the experimental groups. The authors employed the structural connectome obtained from 35 Healthy subjects as basis for many of their analysis. However, starting from the first part of results (pag.4) it is clear that some of the measures employed (Integration, Segregation, and mean FCD, are more similar across CNT and R than between W and CNT. I had the impression that these measures are generally affected by inter-individual variability at a bigger extent that can be controlled. These differences are not emphasized by the authors, but I think they are crucial for the definition of a model that want to find neural correlates of changes in consciousness.

MCS and UWS have different etiologies. Can this affect the pattern of alterations?

Suggested Revisions

Authors stated in the text: line 105: " For the average segregation we observed the opposite pattern [compared to integration]", but the pattern is not the opposite!

100-107: Too many concepts hastily introduced together with a brief description of the corresponding results. Integration and segregation concepts are here presented for the first time but without any prior contextualization. Technical details such as largest subcomponent, modularity and community detection are instead added without explanations of their meaning. Moreover, these technical choices are not supported by an appropriate discussion and leave the results not fully interpretable. E.g., using the size of the largest connected component as a measure of amount of integration in a network is arbitrary and without any further motivation, the choice and, consequently, the results remain arguable. (See also the comment to lines 167-191 and to lines 442-464 below). Line 101: to which time window the authors refer to? Is the same used for FCD analysis? The link to the Methods section (lines 431-434) does not clarify which T is used.

115-122: References to the method used are imprecise and the results are barely discussed. As stated by the authors, claims regarding the recurrence of neural activity are based on the mean of cosine dissimilarity distributions. However, the distributions reported in Figure S1 (which show only a subject) seems to depend not only on the mean, which is clearly not the best informative parameter, but also on other moments. This is also perceivable from panel g) of Figure 1. It is also unclear what the authors mean with "correlation between phase difference matrices..." in the legend of Figure S1 since in the Methods section is stated that the FCD measure was obtained through the cosine dissimilarity.

123-125: Claims are again not supported by enough evidence. Here the authors refer to the graph measures of integration and segregation without giving any detail besides the average values of Figure 1. What "diverse" here means? If connected refers to the network measures and recurrent to the FCD ones is unclear to what "diverse" refers to.

129-132: The authors introduce the Hopf dynamics and the related concept of bifurcation abruptly. To readers non familiar with theory of dynamical system this would result as incomprehensible and obscure and, a consequence, also the related results.

146-157: With the details given here and in Figure 2 interpretation of results is not clear. Reference to the supplementary figures in this paragraph is highly confused and confusing since the metrics to which they refer to (FC and metastability) are not introduced elsewhere in the text and there is not a corresponding section in the Methods. Moreover, is not clear what is the relevance of this paragraph with respect to the line followed by the authors in presenting their results. Statements of lines 151-157 should be explained and supported more in depth: what is the link between the coupling strength g and the "propagation of activity within the network [...] correlation between nodes indirectly coupled"? The same holds for Figure 2 panel c): besides the fact that only a participant is shown, the interpretation of the trend of g should be added. What does the fact that the K-S distance saturates for high g for both MCS and UWS conditions (for both subjects showed) means? How did the authors choose the optimal g ? According to the Methods section (lines 510-511) to a minimum criterion: this is unclear since for both blue curves in Figure 2 (CNT and W subject group) similar minima of the K-S distance was reached for different values

of g . This has also implications on the discussion of panels a and b of Figure 4 .

167-191: This whole section is confused. The concept of network hub is introduced (starting from the title) without giving an appropriate mathematical definition neither here nor in the corresponding Methods section (lines 539-558). The only definition given (line 543) is vague and not rigorous and therefore not reliable and usable (the same holds for legend of Figure 5). How hubs were selected? No degree distribution- based criterion is shown. Moreover, the authors jump from the concept of strength to that of degree. Did the authors compute the rich-club coefficient between hubs of node strength? Regarding the threshold applied to the SC matrices: the strategy followed by the authors implies that a different threshold is chosen for each network. This choice is extremely prone to statistical noise. For this reason, more rigorous thresholding procedures have been proposed and successfully applied on brain networks at different scales (Bordier et al., 2017, *Front. Neurosci.*; Nicoli et al., 2017, *Neuroimage*; Vlasov & Bifone, 2017, *Sci. Rep.*; Mastrandrea et al., 2017 *Sci. Rep.*, Del Ferraro et al., 2018 *Nat. Comm.*, Bardella et al., 2020). Furthermore, the authors did not compare any of the empirical graph-based measures used with a null model. In this way is not possible to state that the identified features are not attributable to low-order constraints (i.e., the weight distribution) and that are not reproduced by a randomization. This is especially the case of the rich-club coefficient which it is known to be strongly dependent on the degree distribution. Therefore, a rigorous estimate of this measure (as all the others presented) should include a comparison with a randomized counterpart (Colizza et al., 2006, *Nat Physics*; Opsahl et al., 2008, *PRL*). The section refers to Figure 3 which is, again, non-descriptive of the content of the results. It is also not possible to deduce, both from the text and the caption, how the panels showed are obtained. Do the show subject-wise or averaged measures? The same comments hold for Figure 4.

194-198: Unclear. What this statement means?

224: The statement is not supported by evidence. How the authors could establish the cause-effect relationship between the results showed and the loss of consciousness? How they demonstrated that the processes showed "lead to loss of consciousness"?

246-255: Unclear how this relates to line of discussion of the results followed until line 245.

320-330: As previously done for the notion of network hubs, the authors discuss part of their results in terms of the core-periphery architecture without giving any quantitative measurement and formal description of this concept, neither in the Results nor in the Method sections.

Appropriate references are also lacking. The term is also inopportunistly used in the Abstract.

409: Usage of a non-uniform notation for the structural connectivity matrices, which have different notations through the manuscript.

442-464: The concepts of network Integration and Segregation which the authors refer to in the main text, are not here explained properly. An adequate discussion in terms of the existing literature is also lacking, leaving authors' choice arbitrary. The only reference provided is an Opinion Article which itself does not include any formal description of the methods used in this manuscript. From what stated, it is deducible that the Integration value was estimated via a percolation process on time-averaged phase-integration matrix. If it is so, references to previous works that adopted the same technique on brain networks must be included. Moreover, the procedure adopted in this study, i.e. measuring the size of the largest component of the network, has been proven to be the non-optimal choice and to be highly susceptible to statistical noise. Hence, being flawed, its usage, need to be motivated by a critical discussion. Otherwise, is unserviceable. Indeed, without a comparison with a null model significance of the results cannot be in any way assessed. In this regard, why no plots are shown about the procedure neither here nor in the supplementary materials? The same holds for the choice of using the integral of the (supposed) percolation curve, which is not supported by any discussion. To the same extent, similar comments hold for the Segregation index based on the modularity maximization, which, again, lack appropriate contextualization and motivation. It has been shown that results obtainable from an adequate percolation process correctly reproduce those of community detection algorithms. Thus, the choice of using both needs to be adequately motivated.

482: Figure 1 does not report the structural matrix as written in the text.

485: How the coefficient Beta was chosen?

577: It is not clear what the authors intend with "the weight of the strength of a node". 146:

Table 2: It is not clear how should I read the table.

Reviewer #3 (Remarks to the Author):

The authors here focus on brain functional dynamics in altered states of consciousness and consider how these are constrained by structural connectivity. The data comes from severely brain-injured patients with disorders of consciousness (DoC) and healthy controls undergoing anaesthesia. Given the difficulty in collecting such data and getting these datasets together, the cohort sizes are satisfactory and the analyses carried out informative. The analytical framework is mostly graph theoretical but there is also interesting computational modelling work presented here. The authors focused on connectome hubs and showed that altered functional dynamics possibly result from a global reduction in network interactions. This is underpinned by integrity of the structural connectomes. The article is interesting and timely, the analyses carried out clear and comprehensive. The writing on occasions is over assertive but not assertive enough when it comes to presenting the novelty in these findings. I have some minor comments which I would like the authors to address before I can recommend the article for publication.

Line 80: What is "the local state of nodes"? The term seems vague.

Line 126: The title needs attention – some section titles could be improved.

Line 204: The terms effective local parameter and effective local dynamics need to be clarified. The term effective has been used in the context of connectivity by Friston et al., 1993, to mean influence of one neural system over another. Do the authors allude to a similar concept here?

Figure 5: Why are the blue areas asymmetrical across L/R hemispheres if these are healthy controls? Why are some of these ROIs in white matter and others in gray matter?

Line 376: CRS-R of DoC patients can be fluctuating. Please provide an indication of how near the functional scans the CRS-R scores were obtained.

Line 546: I am not entirely clear whether 20% is the threshold used to assign a node the role of a hub.

Line 567: Please clarify to whom a request should be addressed to if one wants to work with the data presented here.

Any translational value in these findings?

Supplementary materials:

Not entirely clear whether the healthy controls used for comparisons to DoC patients were scanned with the same MRI parameters as the DoC patients.

DoC patients have usually large lesions. How did the authors deal with this issue in terms of pre-processing the data spatially and temporally? If for example an ROI falls entirely within a lesion, what signal did the authors use in their analysis. Also, lesions can result in distortions following non-linear spatial normalization. How did the authors deal with this? Please provide some examples in supplementary materials.

Reviewer #4 (Remarks to the Author):

Reviewer comments: "Loss of consciousness reduces the stability of brain hubs and the heterogeneity of brain dynamics" by López-González et al.

Summary

This manuscript describes a comparison between fMRI brain dynamics in healthy, wakeful participants and patients in low-level states of unconsciousness, such as disorders of consciousness or under anaesthesia. The authors show that both pathological and pharmacological low-level states of consciousness presented altered network interactions and more homogenous and anatomically-constrained local dynamics than conscious states.

Overall impression

This study appears to support existing findings on network interactions associated with loss of consciousness. The tripartite conclusion of altered network interactions with more homogenous

and structurally-constrained local dynamics, and less stability has validity through impressive statistical analysis. However, the study's novelty must be clarified, as well as a stronger claim for the impact on consciousness as a cause of or a consequence of the observations.

Specific comments with recommendations

In general, the paper is well thought-out with specific methodological contributions to the study of consciousness. The paper should be read for clarity, especially for run-on sentences and use of scientific jargon that does not always convey meaning (line 45, line 82, and throughout the discussion, etc.). Consider rephrasing and tightening common phrases, "we were interested ..." with stronger language (i.e., "we investigated ..." or "we analyzed...").

Abstract

1. What is the difference between structural and dynamical in the abstract (and elsewhere)?
2. Similarly, please clarify about what you mean by "altered dynamics" especially in relation to changes in "local dynamics."
3. I suggest adding the number of participants into your abstract.

Introduction:

4. The introduction would be a good place to highlight what makes this study a novel contribution to the field. There has been a lot of work on coma, consciousness, and how they relate to anesthesia, and your study's innovations were not conveyed clearly here.
5. It is also advisable to clarify terms such as "low-level consciousness" and "awareness" or "arousal." The distinction between arousal/level and content may be helpful here as described in anesthesia-induced loss of consciousness (see PMID 29937348; 28676745). How do those terms relate to your categories of consciousness?
6. "Deep sedation," should be defined, even though the ASA provides a clinical definition. In the discussion, I would recommend the authors to rephrase the term "deep anesthesia." Is this deep sedation, or general anesthesia? "Deep anesthesia" is too vague and conflates analgesia, dose of hypnotic, movement, hemodynamic changes, amnesia, and consciousness. The fields of mechanisms of anesthesia is doing away with the concept of "deep" (PMID: 22011712).

Results

7. Again, be sure to defined "conscious wakefulness".
8. Also consider defining "synchronization dynamics" or "phase integration" in the Introduction.

Discussion

9. Due to differences in anesthetics and their effects on level of sedation, references to "anesthesia" in this manuscript should be changed to "propofol anesthesia." A mention of the effect of different anesthetics would be an important addition to the limitations section (see, i.e., PMID 18988836).
10. There are many references to this study as "consistent" and "in line with" and "supports" previous studies. It is important to emphasize its novelty as well.
11. Many brain regions are involved in loss of consciousness, as you mention. If anesthetics primarily affect the cortex or the thalamus, or brainstem dysfunction, could the results you are seeing be a consequence or a cause of the loss of consciousness? How does the literature of thalamo-cortical or brainstem dysfunction (at the pontine level. See papers by Devor, Franks, Nir and others) relate to your findings?
12. The gender imbalance in several of your study groups (such as the anesthesia group) should be mentioned as a limitation.

Methods

13. Rewrite line 363 for clarity.
14. Did you analyze the DOC patients based on their underlying condition?
15. More details about the anesthetic procedure are necessary (apart from the reference in line 391 to footnote 60). For example, what time after anesthetic administration are recovery levels observed? Why are there so many female participants? What was the anesthetic dose?
16. It would have been preferable to do a concurrent EEG to find correlates with fMRI findings and to determine more accurately the level of sedation.

We thank the reviewers and the editor for their helpful feedback. In the present letter, we address point by point the editor's and reviewers' comments (in blue and italics).

At the end of the document, the updated figures have been included, with a brief explanation of the changes and revised captions, in case it was needed.

Reviewers' comments:

Reviewer #1 (Remarks to the Author):

1 Manuscript overview

The authors present a study of changes in global properties of phase-dynamics in fMRI data sets and model-based from patients that suffered brain injuries leading to a disorder of consciousness and from subjects undergoing propofol-induced anaesthesia. Their study is focused on studying how structural, dynamical, local and network brain properties interplay in the different levels of consciousness. The authors claim that pathological and pharmacological low-level states of consciousness displayed less recurrent, less diverse, less connected, and more segregated synchronization patterns than conscious states.

The authors study these effects using whole-brain models built on healthy and injured connectomes, showing that altered dynamics arise from a global reduction of network interactions, together with more homogeneous and more structurally constrained local dynamics. These effects were accentuated using injured connectomes.

In my opinion, the work is interesting, well written, and presents enough scientific merit. However, the current state of the manuscript presents different aspects that can improve. In my opinion, the work needs to address major and minor issues before further consideration.

We thank the Reviewer for the encouraging impressions/comments on our work.

2 Major

• In my opinion the title of the article does not summarize the main findings. The hubs are not well emphasized in the text as the definition only comes in the legend of figure 5. Additionally, it is not clear to me that as you used it is a genuine topological feature. On the other hand, the stability analysis only appears at the end of the supplementary material.

We thank the reviewer for this remark. As the reviewer pointed out, the stability analysis is one of the main findings reported in the manuscript and this analysis is crucial for understanding the study's contribution to the literature by clarifying the role of the hubs' dynamics in the interaction of the whole network.

In the previous manuscript, we decided to move the bulk of this analysis into the supplemental material due to its mathematical aspect, keeping in mind the goal to satisfy readability for a "broader audience" criteria. In this revised version, following the recommendation of the reviewers we have reconsidered the organization of the manuscript and we have moved the stability analysis and the corresponding methods and figure from the supplemental material to the main text. Moreover, we have also unified the equation's notations to avoid misinterpretations.

Furthermore, we have edited the structural connectivity analysis and clarified the definition and relevance of the structural hubs earlier in the text. While the main part of the manuscript is dedicated to the study and characterization of dynamical properties, we understand the connection to the

underlying structural features is a very important relation to be highlighted. We hope the message we aimed at delivering in the title is now more clear within the text.

• Again in the same line. The article refers 28 times to the supplementary material and important results including one on the title (stability) is in the supplementary material. I believe that either the title should be changed or the article should emphasize results in the main text. Otherwise should balance and reorganize the main text to include what you believe is the most important finding. Figures one and three summarize what you are really doing in this article that is to study using fMRI data sets and model-based data changes in local and global properties of phase-dynamics induced by loss of consciousness.

We have simplified and clarified the text. As we mentioned in the previous point, the manuscript organization has been modified to simplify and make the read easier to understand. The stability analysis and two figures (Figure 3 g and Figure 4) of the supplemental information have been moved to the main text and the analysis of the structural connectivities has also been clarified. Furthermore, we have clarified the definition and the role of hubs in this context (Lines 88-89, 203, 278, 29-295). We hope the flow of the manuscript is now clearer.

• At the end of the introduction the authors refer to topological aspects, however the definition of Hubs as nodes with high strengths is not topological (derived from an adjacency matrix). While later in the text seems that methodologically you end up binarizing the SC still seems to me that you are not referring to a topological property.

The word “topology” is often used in the literature of complex networks simply as a synonym for “structure” or “architecture,” instead of referring to the more formal definition considered in “topology” as a field of mathematics. Our phrasing was following the first approach. We removed the word “topology” from the sentence which, anyway, leaves its meaning unaltered (Lines 86-90).

• There is a mismatch in the definition of strength in line 201 and 536 and what you consider in the linear stability analysis in section 1.4. You consider in the definition of strength the parameter G or not?

We thank the reviewer for pointing this out. It was indeed a mistake. We have defined consistently the strength as the sum of the connections given by the SC, without considering the global coupling g . Furthermore, we have unified the notation using g (lower case) for the coupling strength in the whole work.

• There exist several alternative whole-brain models to study the brain. The Jansen & Rit model and Dynamic Mean-Field are among them. Alternative modeling approaches can be used to study Integration, Segregation, and Mean FCD. I'm not asking for further analysis but the article should explain more clearly why you are focusing on Stuart Landau oscillators rather than more biophysically grounded alternatives.

We have added discussion with further references. As the reviewer indicates, there exist alternative whole-brain models to study brain dynamics. Generally, those models are described depending on the dynamics occurring at the node level, i.e. the brain region's intrinsic behaviour. One can find in the literature, among others, conductance-based biophysical models (Breakspear et al., 2003, Honey et al., 2007), oscillatory dynamics considering the population of neurons composed of excitatory and inhibitory neurons (Ghosh et al., 2008, Deco et al., 2009), local oscillators (Kuramoto (Cabral et al., 2011) and Stuart Landau oscillators (Deco et al., 2017)), attractor network of spiking neurons (Deco et al., 2012) or noise diffusion network (Gilson et al., 2016). Our primary goal in this study was to study

a whole-brain model that is able to produce oscillations, as needed to model the synchronization statistics of the data. Indeed, phase-synchronization analysis allows describing the global dynamics and correlations at each time-point thus avoiding sliding windows (Ponce-Alvarez et al., 2015). Intending to describe the phase dynamics of each brain state, we chose a whole-brain model based on oscillators. In particular, we chose the Stuart Landau oscillators, which represents the normal form of a Hopf bifurcation, i.e., the universal behaviour around a bifurcation producing oscillation through a limit-cycle. Despite its simplicity and non-biological origin, the model has shown to generate rather an accurate fit to the BOLD dynamics, beyond the success of other models in the past (Deco et al., 2017, Saenger et al. 2017, Jobst et al, 2017). We have included a paragraph in the discussion where we justify the model applied here (Lines 467-475).

The reviewer also asks for a justification of the integration/segregation measures used in this study that will be explained and justified in the next point.

- *There exist alternative definitions of Integration and Segregation. This fact should be emphasized and the choice better motivated.*

As far as we can tell, integration and segregation are concepts that are still relatively elusive towards a strict definition for several reasons, even at the conceptual level. From a quantitative point of view, at this moment, we are unaware of other metrics which can reliably indicate integration or segregation and which can be quantified out of functional connectivity data. If the Referee provided some concrete recommendations, we will be pleased to consider them.

- *I can understand that some sensitive data cannot be shared. However, in my opinion, the code used to obtain the results should always be available, readable, and usable without even need to request it.*

We will make the code publicly available through GitHub. This will be specified in the “Code Availability” section.

Furthermore, part of the preprocessed dataset will be available in EBRAINS (the platform being built by the Human Brain Project) in the near future.

3 Minor

- *In formula 4 there is a cos missing, and I will recommend to use kxk instead of $| \cdot |$*

We have re-written the formula accordingly (Line 611).

- *The formula of angle ... dot product or internal product.*

As we indicated in another answer before we have unified notations, so the confusion among notations is avoided.

- *The authors used a unidimensional notion of consciousness, i.e. states can be mapped from low-levels to high-levels of consciousness, but there are some other perspectives that argue that consciousness is better characterized as a multidimensional construct (see <https://www.ncbi.nlm.nih.gov/pmc/articles/PMC7167214/>). Even when the results show that different states can be mapped onto a single dimension, a small discussion on how their results relate (or not) to a multi-dimensional account for consciousness could highly enrich the article.*

This is an interesting remark. We added this point to the discussion. As correctly addressed by the reviewer, in this study, we investigated the brain dynamics and mechanism of low-level states of

consciousness in comparison to conscious wakefulness. Low-level states of consciousness were described based on the two main dimensions proposed to explain consciousness, i.e. awareness and wakefulness (Laureys, S et al. 2015). We see the point of the reviewer to describe consciousness taking into account other brain states characterized by altered consciousness, such as under psychedelic drugs or meditation, where the content of conscious processing is increased, and in order to define the state of consciousness, other dimensions apart from wakefulness and awareness have to be taken into account (such as visual perception, cognition or/and experience of unity). As we believe that this framework could be applied to other brain states with altered consciousness (Lines 458-465).

- *The selection of colors in the boxplots is not appropriate to see the values of all the subjects, specially in the red and the blue boxes.*

We have changed the images adding more transparency to the boxplot, so each subject's point is properly visible now (see figures 1 and 2 in the main text and S3 in the supplemental material). See the updated figures in the final part of this document.

- *Formula 1 in the supplementary material is t going from 1..T?*

Thank you. We corrected the mistakes in equation 1 and 2 of the supplementary material. We hope that with the new changes these equations are now acceptable.

- *Figure S6 should consider a line and an inset with the p values in each subplot.*

We have considered the reviewer's suggestion, and we have re-edited the figure. Moreover, considering the reviewers' comments about the many references to the supplemental, previous Figure S6 has been moved to the main text (added to Figure 3g). See the updated figures in the final part of this document.

- *In my estimation you can do better using the information of the tables S1 and S2, probably a good idea would be to compare the ROIs with highest absolute difference in bifurcation parameter between DOC and anaesthesia as it seems to have an interesting overlap.*

We agree with the reviewer that comparing the ROIs in those tables could show an interesting overlap, but we believe that this is not the aim of the tables. Tables S1 and S2 are the detailed information of Figure 3f, where the areas with the most notable differences in the bifurcation parameters are depicted. Those tables reflect the exact areas, i.e. names and correspondence with AAL labels, where we found the maximum difference in the bifurcation parameters. We believe that the brains depicted in Figure 3f are already reflecting the overlap of the areas in the tables.

- *Considering that you are counting the number of nodes within the largest component I would refer to the size of the largest component not the length.*

We revised and corrected the nomenclature everywhere.

- *Line 134. The name of the atlas could be mentioned in the results for clarity.*

We have added more information about the atlas and reference. (Line 159-160)

- *I only know graphs, subgraphs and hypergraphs. What is a super-graph?*

Indeed. We ignore how this term ended in the manuscript, most likely due to a typo. We apologise for this.

- *line 365 Liège.*

We have corrected the accent mark. (Line 501)

- *line 424 $z(t) = s(t) + i \cdot H[s(t)]$*

We have written the equation properly. (Line 564)

- *There is a typo on line 471.*

We have corrected the typo. (Line 608)

• *Discussion, line 317. The authors argue that the reduced variability and increased stability of hubs are essential for consciousness. However, modeling and empirical results show that in the psychedelic state many brain hubs increase their entropy (i.e. increased variability, see for example <https://www.nature.com/articles/s41598-020-74060-6>). During the psychedelic state consciousness is not lost, rather, conscious content seems to increase, so, how can these results on psychedelic states could be integrated in the framework presented by the authors? Is the psychedelic state an example where the stability of hubs is reduced, but consciousness is preserved ?*

This is a very interesting question. We agree with the reviewer's hypothesis. According to previous work, during psychedelic states, the brain entropy increases in an information-theoretic notion (Schartn et al. 2016) and considering the local firing rates (Herzog et al. 2020) alongside the conscious content. In agreement with the reviewer's statement, we conjecture that during psychedelic states, the hubs will lose stability, and this will allow the brain to explore a more extensive repertoire of states that leads to an increase of conscious content and perception, among others. This is contrary to what we observed in the low-level states of consciousness presented in this study. Furthermore, in conscious wakefulness, the hub's dynamics ensure the network's stability compared with the low-level states of consciousness. As we found very interesting this point, we have included a short discussion about this (Line 461-465).

• *Discussion. Line 354. Any ideas on how anesthesia and DOC states converge onto similar dynamics despite their divergences on the underlying physiological mechanisms? A small discussion on the possible physiological mechanisms that make these two unconscious states similar in terms of the departure from conscious wakefulness dynamics would greatly improve the reach of the article. See for example [https://www.cell.com/trends/cognitive-sciences/fulltext/S1364-6613\(20\)30175-3](https://www.cell.com/trends/cognitive-sciences/fulltext/S1364-6613(20)30175-3).*

Indeed, although the underlying physiology for unconsciousness differs between DOC patients and general anaesthesia, the dynamics at the whole-brain level seem to be similar. Indeed, the relatively similar phenomenology (Brown, E. N. et al., 2010) of the two different states might have a shared cellular basis for the observed global dynamics (Aru J. et al., 2020). Layer 5 pyramidal neurons consist of three compartments, an apical compartment that integrates information from higher-order processes, the basal compartment receiving information from lower compartments, and the coupling compartment that integrates activity from both pyramidal compartments and the thalamus. In DOC patients, this circuit might be predominantly affected by the thalamus and/or the basal and coupling compartments while in anaesthesia, the apical compartment seems most affected. In all these situations, global dynamics would be hampered as observed in our current study. Previous empirical evidence for the similar global dynamics in both pathological and pharmacological (as well as physiological) states of unconsciousness exist, the perturbational complexity index can reliably distinguish levels of (un)consciousness regardless of the cause. Thus, interactions and global

behaviours in conscious wakefulness differ from low-level states of consciousness, yet show similar patterns in DOC and propofol anaesthesia, even if consciousness has been reduced or lost by different underlying physiological mechanisms.

It is important to point out that the local coupling parameters are differently distributed in UWS patients and sedation, the former showing a predominant effect of the precuneus and posterior cingulate and the latter being more affected in the frontal cortex concordant with propofol's site of action (Brown, E. N. et al., 2010). As the thorough investigation of local changes is out of the scope of the current paper, no formal statistical comparison between UWS and sedation were performed.

We have included a brief discussion in the Conclusions (Lines 450-452).

• *In the supp material section 1.4 you refer to Figure S6 when should refer to Figure S8*

The mistake has been corrected, but it should be noted that this figure has been moved to the main text so the reference has changed. See the updated figures in the final part of this document.

Reviewer #2 (Remarks to the Author):

The authors investigated fMRI brain dynamics from patients showing disorder of consciousness following brain injuries and from subjects showing altered consciousness due to anaesthesia. The altered states of consciousness observed were different from not altered states of consciousness. Starting from these observations they declare that developed models (homogenous and heterogenous) to investigate the mechanisms (in terms of global and local dynamics) underlying the loss of consciousness. The authors interpreted their results in a whole-brain model framework constrained by a structural connectome. Despite the idea is by itself interesting, some methodologies are vaguely presented, contain inaccuracies and an appropriate critical discussion in terms of other works with same techniques is missing. This is especially true for the graph theory and complex networks analysis tools. In what follows are the details. The topic that the authors declare to address is of general interest in the field of the biology. However, the main goal of the paper in not convincingly reached in my opinion.

I have some general comments regarding the paper and some suggested revisions:

We thank the comments and the suggestions of the reviewer. We have substantially revised the manuscript and we hope the revised version is more clear and readable.

General comments:

In general, I think that the paper is too dense and rich of analysis that are not well explained, and gives the impression that the main statements lack of scientific clarity. Results presentation is not accurate and self-consistent, and some logical leaps are present. The frequent references both to the Methods section and to the supplementary material make difficult to follow the author's claims precisely and generate confusion. Furthermore, the continuous references to many of the supplementary figures leave the doubt that many of them are be crucial to properly understand the results. A non-expert reader would not be able to disentangle the bundle of information reported.

Considering the comments of the reviewers about the confusion of the references to the supplementary material and methods, we moved some of the analysis previously presented in the supplemental material to the main text: the stability analysis method explanation and two figures (in the revised manuscript Figure 3g and Figure 4). See the updated figures in the final part of this document. Our main goal with these changes is to clarify the flow of the manuscript and to strengthen the analysis presented. Please note that such decisions were taken in the initial manuscript as an

attempt to balance the readability for the broader audience but we understand the concerns of the reviewers in this respect. In this revised version, we hope that all information and analysis needed to understand the contribution of the study are presented in the main text.

This is also the case of the Figures: the captions are often confused and not well planned in terms of content. These aspects require the reader to jump continuously affecting its ability to focus on the main messages. Furthermore, I found out that some of the important concepts (and measure and indexes) are not clearly defined (f.e. the concept of Hubs that appears in the title), and or the definition of the employment of a certain measure is not always justified. Some sections of the Methods are imprecise and hasty, especially those regarding graph theory tools.

Thank you. The methods have been revised and simplified. The captions have also been revised (See the updated figures in the final part of this document). Conceptual explanations regarding the methodology have been improved along with the text. Also, the last section dedicated to the network analysis of the structural connectivity has been re-written. In particular, we replaced the results with weighted network analysis to avoid the binarization of the SC matrices. Please see more details below.

I have a main concern regarding the selection of the experimental groups. The authors employed the structural connectome obtained from 35 Healthy subjects as a basis for many of their analysis. However, starting from the first part of results (pag.4) it is clear that some of the measures employed (Integration, Segregation, and mean FCD, are more similar across CNT and R than between W and CNT. I had the impression that these measures are generally affected by inter-individual variability at a bigger extent that can be controlled. These differences are not emphasized by the authors, but I think they are crucial for the definition of a model that want to find neural correlates of changes in consciousness.

We agree with the reviewer that this is a crucial point in this type of analysis. Indeed, we are currently working in this line to improve the interpretation of our results and, in particular, in the cases where the application can be made in a clinical context, where the individualized analysis is key to understanding the pathological neural mechanisms. In particular, in DOC patients, each patient has a unique lesion, which affects different brain areas.

We consider that the analysis performed in this study is the first step towards this goal. We have focused on the average alterations at the group level without considering the inter-variability within the groups in order to reveal the common factors leading to disorders of consciousness or unconsciousness in sedation. Once we understand these common grounds and the associated brain mechanism, we believe that it will be easier to study individual subjects and cross-subject variability.

MCS and UWS have different etiologies. Can this affect the pattern of alterations?

We see the concern of the reviewer in relation to the etiologies affecting the patients of each group. Here, in this study, our aim was to perform a population average study of each group, independently of the etiology, and considering the behavioural scans (CSR-R) to classify the groups. Hopefully, in the future, we will have larger datasets where we will be able to perform individual analysis of each subject, investigating subjects in the same groups but with different etiologies. However, we have studied the effect of the etiology considering the two patient groups (MCS and UWS). We did the analysis based on DOC patients' etiology classified into two groups, that is traumatic brain injury (i.e., TBI) and nonTBI (e.g., anoxia, hemorrhage, infection) patients. Due to limited sample sizes of the less frequent etiologies a more fine-grained distinction could not be made. We did not find any significant difference for the empirical measures and whole-brain modelling measures (i.e. global coupling

strength) between the TBI and nonTBI patient groups. This indicates that, although the test might be underpowered at this stage with relatively low sample sizes, the altered brain network dynamics we observed are sensitive to consciousness levels and independent from the patient's underlying condition. The statistics were computed by using a two-sample *t*-test.

Integration: $p=0.3049$

Segregation: $p=0.0160$

Phase-interaction fluctuations: $p=0.2674$

mean FCD: $p=0.2835$

Global coupling g : $p=0.9371$

Suggested Revisions

Authors stated in the text: line 105: " For the average segregation we observed the opposite pattern [compared to integration]", but the pattern is not the opposite!

With this sentence, we wanted to highlight the fact that the tendency of the integration and segregation are opposite in the sense that the integration decreases for low-level states of consciousness compared to wakefulness while segregation increases for low-level states of consciousness. Considering the point of the reviewer, we have rewritten the sentence in order to make it clearer and avoid misinterpretations. In this new version, we have indicated expressly the tendency for segregation and the comparisons that were significant (Line 119-121).

100-107: Too many concepts hastily introduced together with a brief description of the corresponding results. Integration and segregation concepts are here presented for the first time but without any prior contextualization. Technical details such as largest subcomponent, modularity and community detection are instead added without explanations of their meaning. Moreover, these technical choices are not supported by an appropriate discussion and leave the results not fully interpretable. E.g., using the size of the largest connected component as a measure of amount of integration in a network is arbitrary and without any further motivation, the choice and, consequently, the results remain arguable. (See also the comment to lines 167-191 and to lines 442-464 below).

We have rephrased and extended the explanations in this section. See the renewed paragraphs in the main text. It is now hopefully more informative, readable and accessible. The reviewer refers to the measures of integration and segregation several times. Please find more concrete answers in the responses below. At this point, however, we should confirm that the sentence stating "*Integration was measured as the size of the largest subcomponent*" was an inaccurate explanation and has been corrected.

Line 101: to which time window the authors refer to? Is the same used for FCD analysis? The link to the Methods section (lines 431-434) does not clarify which T is used.

We have clarified that the integration and segregation are computed from the time-averaged phase-interaction matrix, without applying any specific time window as is the case of the FCD. In the methods, we have included the definition of the time-averaged phase-interaction matrix and how it is computed averaged across time.

115-122: References to the method used are imprecise and the results are barely discussed. As stated by the authors, claims regarding the recurrence of neural activity are based on the mean of cosine dissimilarity distributions. However, the distributions reported in Figure S1 (which show only a

subject) seems to depend not only on the mean, which is clearly not the best informative parameter, but also on other moments. This is also perceivable from panel g) of Figure 1.

It is also unclear what the authors mean with “correlation between phase difference matrices...” in the legend of Figure S1 since in the Methods section is stated that the FCD measure was obtained through the cosine dissimilarity.

Firstly, we want to indicate that panel a of Figure S1 corresponds to one subject, but panel b corresponds to the normalized distributions of all the values of the FCD matrices, of all the subjects in each group. We have corrected the caption to make it more straightforward. We also see the reviewer's point that the FCD patterns can be described by other statistical parameters apart from the mean. Considering the different analysis performed through the study, the mean was chosen for simplicity, and we did not go into details to analyze those matrices, even if we consider that could be interesting further research. Indeed, we understand that a full analysis of the spatio-temporal patterns and their recurrence, as compared across the states of consciousness, would be a very interesting topic. However, we believe that such a deeper analysis could be the aim for a stand-alone manuscript, and thus beyond the possibilities of the present work. Please notice that in the present manuscript, our intention is to keep this analysis simplified before diving into the modelling part, which is the main novelty and rather dense.

We would also like to note that there was a mistake in the legend of Figure S1 where we wrote that the FCD was calculated based on the correlation among matrices but as we explained in the methods, the FCD was computed by calculating the cosine similarity. The caption of Figure S1 has been corrected. Thank you for pointing it out.

123-125: Claims are again not supported by enough evidence. Here the authors refer to the graph measures of integration and segregation without giving any detail besides the average values of Figure 1. What “diverse” here means? If connected refers to the network measures and recurrent to the FCD ones is unclear to what “diverse” refers to.

Please find in answers below the clarification about measures for segregation and integration.

The term ‘diverse’ has been rephrased and explained the meaning by describing the level of fluctuations in the phase-locking matrices.

129-132: The authors introduce the Hopf dynamics and the related concept of bifurcation abruptly. To readers non familiar with the theory of dynamical systems this would result as incomprehensible and obscure and, a consequence, also the related results.

We understand the point of the reviewer. We have now enhanced the beginning of the modelling part to provide a more descriptive and smoother introduction of the model (Lines 148-156). We hope this makes the section more accessible.

146-157: With the details given here and in Figure 2 interpretation of results is not clear. Reference to the supplementary figures in this paragraph is highly confused and confusing since the metrics to which they refer to (FC and metastability) are not introduced elsewhere in the text and there is not a corresponding section in the Methods. Moreover, is not clear what is the relevance of this paragraph with respect to the line followed by the authors in presenting their results. Statements of lines 151-157 should be explained and supported more in depth: what is the link between the coupling strength g and the “propagation of activity within the network [...] correlation between nodes indirectly coupled”?

Thank you for this important remark that was not fully explained in the MS. The supplementary figure that is referred in this paragraph aims to show that the best fitting of the model is not only achieved by

the minimal distance between the distribution of the empirical and simulated distribution of the FCD, but that other statistics that have been previously suggested to account for the spatiotemporal properties of the signals such as the Pearson correlation FC matrix or the phase-interaction fluctuations which has been shown in Figure 1. We have corrected the metastability term that, as the reviewer pointed out, has not been defined neither in the main text nor in the supplementary methods (it is referred to as the phase-interaction fluctuations). The Pearson correlation FC is defined in the supplemental information with detailed information about its calculation.

On the other hand, following the reviewer suggestion, we have included a brief explanation of the global coupling strength, g , highlighting the role of the global coupling in explaining the global conductivity of the brain network (Lines 184-191). Indeed, the global coupling g explains the implication of the links in the simulation of the spatiotemporal properties of brain dynamics.

The same holds for Figure 2 panel c): besides the fact that only a participant is shown, the interpretation of the trend of g should be added. What does the fact that the K-S distance saturates for high g for both MCS and UWS conditions (for both subjects showed) means?

How did the authors choose the optimal g ? According to the Methods section (lines 510-511) to a minimum criterion: this is unclear since for both blue curves in Figure 2 (CNT and W subject group) similar minima of the K-S distance was reached for different values of g . This has also implications on the discussion of panels a and b of Figure 4 .

The curves shown in Figure 2 panel c) depict the Kolmogorov-Smirnov distance between the simulated and empirical FCD for different values of global coupling g . So, we run the model with different values of g and from the simulated time series we can calculate the FCD, in the same manner as done for the empirical time series (explained in the Methods). Then, for each g , we compare the FCD distributions.

In the curves that are shown in the manuscript, especially for the MCS and UWS examples, for low and large g , the curves of the KS distance saturates, i.e. the distance between the mean of the distributions show large values. For a specific range of g , the curves show low values, which suggest that the means of both distributions are closer. This range is the range that we consider that the fitting of the model is the best to simulate the empirical dynamics. For simplicity, we have considered the minimal value of the curves as the optimal g , even if we are aware that the best fitting could be done for a wider range than the g where the KS distance has its minimal value, following previous work that successfully simulated brain dynamics of resting-state (Deco et al. 2017).

We thank the reviewer for pointing this out and have included a brief explanation to clarify the results (Lines 171-176).

167-191: This whole section is confused. The concept of network hub is introduced (starting from the title) without giving an appropriate mathematical definition neither here nor in the corresponding Methods section (lines 539-558). The only definition given (line 543) is vague and not rigorous and therefore not reliable and usable (the same holds for legend of Figure 5). How hubs were selected? No degree distribution- based criterion is shown.

Moreover, the authors jump from the concept of strength to that of degree. Did the authors compute the rich-club coefficient between hubs of node strength? Regarding the threshold applied to the SC matrices: the strategy followed by the authors implies that a different threshold is chosen for each network. This choice is extremely prone to statistical noise. For this reason, more rigorous thresholding procedures have been proposed and successfully applied on brain networks at different

scales (Bordier et al., 2017, *Front. Neurosci.*; Nicoli et al., 2017, *Neuroimage*; Vlasov & Bifone, 2017, *Sci. Rep.*; Mastrandrea et al., 2017 *Sci. Rep.*, Del Ferraro et al., 2018 *Nat. Comm.*, Bardella et al., 2020). Furthermore, the authors did not compare any of the empirical graph-based measures used with a null model. In this way it is not possible to state that the identified features are not attributable to low-order constraints (i.e., the weight distribution) and that are not reproduced by a randomization. This is especially the case of the rich-club coefficient which it is known to be strongly dependent on the degree distribution. Therefore, a rigorous estimate of this measure (as all the others presented) should include a comparison with a randomized counterpart (Colizza et al., 2006, *Nat Physics*; Opsahl et al., 2008, *PRL*). The section refers to Figure 3 which is, again, non-descriptive of the content of the results. It is also not possible to deduce, both from the text and the caption, how the panels showed are obtained. Do they show subject-wise or averaged measures? The same comments hold for Figure 4.

We agree with the Referee that this section required revision. Your comments as well as comments from other referees evidenced that this section was poorly aligned with the rest of the manuscript. And we apologise for this. We also admit that changing to a binarised analysis at the end of the manuscript while the whole-brain modelling had been performed on the weighted SC matrices, was incongruent. On the other hand, please notice that the main goal of the paper is the dynamical lessons we learn and the network analysis is not intended to be a complete analysis, but supportive to complement the dynamical results.

In the revised version we have restricted to a weighted network analysis thus avoiding the need (and the troubles) of binarising the SC matrices. Regarding the identification of hubs, well, “a hub is a node with many connections,” that is it. As troublesome as this can be, truth is that several concepts in graph analysis are rather loose and lack a strict mathematical definition. Besides, the notion of hub can be somehow different if our focus is large and sparse networks, such as the internet or the world-wide web, or we study small and dense networks, as brain connectomes.

Our evidence for hub ROIs was (and still is) based on Figure 5b, which compares the node strengths of all ROIs for healthy subjects and patients. It is evident in that plot that the strength of some nodes in the healthy subjects (blue line) are larger than in most of the ROIs. Also, that those ROIs lose their larger strength in the patients. But we agree this was not sufficiently explained. We hope the revised text is clearer. As suggested by the reviewer we also include the distributions of node strengths for the population-averaged connectivity in healthy participants and patients, the new Figure 5d (See the updated figures in the final part of this document). In that plot, we can see that at the range of larger strengths, the distributions in patients rapidly decay while the distribution for controls presents a longer tail. We now use this longer tail to mark the set of structural hubs. This set is practically the same as before, but better justified.

We have also simplified the content related to the rich-club and aligned it better to the message of the paper. Once we have shown that the connectivity of hub ROIs in healthy subjects is very much altered in the patients with DOC, the goal is just to evidence that the interconnections between those ROIs is also affected in the patients. This is somehow trivial because if the hubs are gone, then it is not possible for those networks to have a rich-club. Anyway, to illustrate this more comprehensively we present two new plots. Figure 5f shows the weighted k -density on the population average SC in the three conditions. Clearly, k -density suffers an early cut-off in the patients due to the lack of hubs. Figure 5g, shows the average SC link weight between those ROIs identified as hubs (in the case of healthy subjects) and the average link weight between the rest of the regions. Clearly, the connections between the hub ROIs is notably stronger than the connections between non-hubs, but it largely decreases for the DOC patients.

We hope this revised presentation of the analysis is more explicit and better understood.

194-198: Unclear. What this statement means?

Our goal in this section is to disentangle the local dynamics from the network interaction effects intrinsically affecting the local parameters presented in the previous section, where the local bifurcation parameters were calculated considering the local dynamics of nodes embedded in a network. The idea of defining this effective bifurcation parameter is to study a perturbed coupling of the local dynamics where the effective bifurcation parameter also includes the network effects, and it contains the effect of the local dynamics and the network effects. Following the reviewer's suggestion, we have included a sentence to clarify the rationale behind defining and using effective local bifurcations. (Lines 236-245)

224: The statement is not supported by evidence. How the authors could establish the cause-effect relationship between the results showed and the loss of consciousness? How they demonstrated that the processes showed "lead to loss of consciousness"?

Our goal with this statement was not to establish a causal-effect relationship between the loss of consciousness and results, but to highlight that in this last analysis, we are studying the structural damage measured in DOC patients' connectomes, which is the origin of the loss of consciousness in those participants. But considering the reviewer's comment, and the confusion that could raise, we have removed this sentence.

246-255: Unclear how this relates to line of discussion of the results followed until line 245.

As we indicated in the introduction of the manuscript, this whole-brain model based on Hopf bifurcations allowed the exploration of how three factors interplay to create the brain dynamics underlying each level of consciousness. These are the local dynamics, network effects and the structure of the underlying anatomical connectivity. In the previous sections of the results, we have inspected the effects in the model coming from the local dynamics and the network effects. In this last section, our goal is to investigate the effect in the models of using different structural connectivity, which is the origin of the loss of consciousness.

320-330: As previously done for the notion of network hubs, the authors discuss part of their results in terms of the core-periphery architecture without giving any quantitative measurement and formal description of this concept, neither in the Results nor in the Method sections. Appropriate references are also lacking. The term is also inopportunistly used in the Abstract.

We admit that the term "core-periphery" is introduced implicitly, without an explicit definition. We expected the target audience would be familiar with it. Nevertheless, we have done some refinements in this respect. In particular, the term was removed from the abstract. The term "core-periphery networks" is now only used in the discussion. In general, we would expect that the average reader that the manuscript targets understand that a hierarchical/modular network with a rich-club is, by definition, a core-periphery architecture. Therefore, the quantification of the structural hubs and rich-club, and the measure of modularity from functional connectivity are quantitative indicators showing the presence of a core-periphery architecture. However, we hope we were more explicit now. The manuscript also cites Zamora-López et al. *Front. Neuroinform.* (2010) and Zamora-López et al. *Front. Neurosci.* (2011) which are the first references demonstrating a core-periphery architecture in cortical networks, and van den Heuvel & Sporns *J. Neurosci.* (2011) identifying such architecture in the human connectome (despite the term was not explicitly used in those early references). While these references account for the anatomical connectivity only, for the dynamical implications we now included two references: Gomez-Gardeñes et al. *PLoS ONE* (2010), and Gollo et al. *Phil.Trans. B* (2016).

409: Usage of a non-uniform notation for the structural connectivity matrices, which have different notations through the manuscript.

We agree that the multiple notations of the structural connectivity matrices (i.e. structural connectomes and structural connectivities) are hindering the proper and comfortable reading of the text. For this reason, and following the reviewer's advice, we have unified the notation and called *structural connectivity* to the SC matrices.

442-464: The concepts of network Integration and Segregation which the authors refer to in the main text, are not here explained properly. An adequate discussion in terms of the existing literature is also lacking, leaving authors' choice arbitrary. The only reference provided is an Opinion Article which itself does not include any formal description of the methods used in this manuscript. From what stated, it is deducible that the Integration value was estimated via a percolation process on time-averaged phase-integration matrix. If it is so, references to previous works that adopted the same technique on brain networks must be included. Moreover, the procedure adopted in this study, i.e. measuring the size of the largest component of the network, has been proven to be the non-optimal choice and to be highly susceptible to statistical noise. Hence, being flawed, its usage, need to be motivated by a critical discussion. Otherwise, is unserviceable. Indeed, without a comparison with a null model significance of the results cannot be in any way assessed. In this regard, why no plots are shown about the procedure neither here nor in the supplementary materials? The same holds for the choice of using the integral of the (supposed) percolation curve, which is not supported by any discussion. To the same extent, similar comments hold for the Segregation index based on the modularity maximization, which, again, lack appropriate contextualization and motivation. It has been shown that results obtainable from an adequate percolation process correctly reproduce those of community detection algorithms. Thus, the choice of using both needs to be adequately motivated.

Following the response to Referee #1 to the same topic, one has to admit that integration and segregation are elusive concepts that, for several reasons, still escape strict definition. From a quantitative point of view, we are unaware of other metrics to faithfully indicate integration or segregation that could be quantified out of functional connectivity data. If the Referee could provide other optimal recommendations, we will be pleased to consider them.

Regarding the metric of integration, we shall clarify that the metric is not just the size of the largest component of a thresholded FC matrix. For this metric of integration, the FC matrix is thresholded across all possible thresholds, from 1 to 0. At each threshold, links smaller than the threshold are discarded and the largest component of the resulting binary matrix is identified. Finally, integration is quantified as the sum of the sizes of the largest components at all thresholds, which allows capturing the hierarchical organization of the functional connectivity across all possible thresholds. Several studies have used this measure of global integration as a measure to distinguish between states or define basic properties of the spatiotemporal dynamics, all of them based on the review, Deco et al. 2015, cited in our study (Deco et al., 2017, Neuron; Deco et al., 2017, eNeuro; Lord et al., 2017, Philos Trans A Math Phys Eng Sci; Cruzat et al., 2018, Neuroimage; Adhikari et al., 2017, Brain; Deco et al., 2018, Neuroimage). We have added some of those references in the main text.

Regarding segregation, its definition is probably a bit less problematic than integration. Segregation refers to the breakdown of a networked dynamical system into functional subcomponents. Community detection methods do precisely that, they aim at identifying elements of the system that are more cohesively connected with each other than they are from others. Modularity is a measure to quantify how good was the job done by the partition algorithm to divide the system into meaningful compartments (or clusters). Therefore, the modularity returned by community detection methods, on a functional connectivity matrix, is a fairly decent indication of how functionally segregated the system is into subcomponents.

Of course, we have enhanced conceptual explanations in the text, which we hope will be clearer and more informative.

482: Figure 1 does not report the structural matrix as written in the text.

Thank you for noticing this error. We have removed this figure that was not giving new information.

485: How the coefficient Beta was chosen?

This coefficient was chosen based on previous studies that applied successfully this model to simulate and obtain information of the mechanism in resting-state, application of DBS and altered states of consciousness, i.e. sleep (Deco G, et al. 2017, Saenger V. et al., 2017, Jobst B et al., 2017). We have included these references in the text to support our choice (Line 627).

577: It is not clear what the authors intend with “the weight of the strength of a node”.

We have rewritten the title to clarify the concepts. (Line 667)

146: Table 2: It is not clear how should I read the table.

This table gives information about the exact values of the average global coupling parameters based on these parameters' values for all subjects depicted in Figure 2d. Those values are explicitly indicated in the table for reproducibility so that the simulations can be repeated with the global coupling strength's exact value.

Reviewer #3 (Remarks to the Author):

The authors here focus on brain functional dynamics in altered states of consciousness and consider how these are constrained by structural connectivity. The data comes from severely brain-injured patients with disorders of consciousness (DoC) and healthy controls undergoing anaesthesia. Given the difficulty in collecting such data and getting these datasets together, the cohort sizes are satisfactory and the analyses carried out informative. The analytical framework is mostly graph theoretical but there is also interesting computational modelling work presented here. The authors focused on connectome hubs and showed that altered functional dynamics possibly result from a global reduction in network interactions. This is underpinned by integrity of the structural connectomes. The article is interesting and timely, the analyses carried out clear and comprehensive.

The writing on occasions is over assertive but not assertive enough when it comes to presenting the novelty in these findings. I have some minor comments which I would like the authors to address before I can recommend the article for publication.

We thank the reviewer for his/her comments. At this point, we would like to clarify that the main objective of the manuscript is to study and characterise the network level dynamical features and alterations under disorders of consciousness. Please note that the graph analysis in this manuscript is intended to be minimal, restricted to build the bridge between structural features – such as the presence of structural hubs – and the dynamical observations.

Line 80: What is “the local state of nodes”? The term seems vague.

By 'local states of the nodes' we refer to the dynamical regime – or dynamical working-point – expressed by the individual ROIs. The network model here employed is based on assuming that each ROI behaves as a noisy oscillator following dynamics governed by the normal form of the Hopf

oscillator, allowing exploring each node's local dynamics. This local model, representing each ROI's activity, allows to transit from stable asynchronous behaviour governed by noisy fluctuations to sustained oscillatory synchronized behaviour.

We have included a brief clarification of what means the local state of the node in the results. We hope that with this brief explanation included in the results and methods, this term can be fully understood.

Line 126: The title needs attention – some section titles could be improved.

According to the comment of the reviewer, we have modified the section titles to clarify the concepts. Modified titles:

- Decreased brain phase dynamics complexity in low-level states of consciousness. (Line 98)
- Decreased global coupling in low-level states of consciousness (Line 144)
- Loss of regional dynamic heterogeneity in low-level states of consciousness (Line 192)
- Disentangling regional and network effects (Line 235)
- Relation between the strength of a node and its dynamics (Methods section)(Line 667)

Line 204: The terms effective local parameter and effective local dynamics need to be clarified. The term effective has been used in the context of connectivity by Friston et al., 1993, to mean influence of one neural system over another. Do the authors allude to a similar concept here?

The term “effective connectivity” as defined for example in Friston et al, 1993 refers to the strength of causal interactions between two brain regions. Here, we used the term “effective” as typically used in Physics and Mathematics. We have defined the effective bifurcation parameters with the goal of disentangling the local dynamics from the network effects. The idea of defining this “effective bifurcation” parameter is to study a perturbed coupling of the local dynamics where the effective bifurcation parameter *accounts for the network effects* so that it contains both the effect of the local dynamics and the network effects.

We see the point of the reviewer and to clarify the confusion this could raise, we have included a sentence that clarifies the origin of the term in this study. (Lines 668-669)

Figure 5: Why are the blue areas asymmetrical across L/R hemispheres if these are healthy controls? Why are some of these ROIs in white matter and others in gray matter?

Even if the structural connectivity matrices derived via tractography are undirected (therefore the matrices are symmetric) this does not imply that the connectivity of homotopic regions in left and right hemispheres is identical. The asymmetries seen in Figure 5 are just the reflection of asymmetries in the connectivity of mirrored ROIs as returned by tractography. Whether this is a consequence of true anatomical differences or simply errors/biases in diffusion imaging and tractography, is hard to tell and could be the subject of a dedicated study. Please note that these results have slightly changed with the new pipeline performed in the structural connectivities (see the updated figures in the final part of this document).

Line 376: CRS-R of DoC patients can be fluctuating. Please provide an indication of how near the functional scans the CRS-R scores were obtained.

5 CRS-R assessments were performed within a period of 14 days, usually within one week. The best CRS-R diagnosis was used for clinical diagnosis. One CRS-R assessment was performed before the MRI acquisition on the same day, yet the clinical diagnosis was made based on the best out of 5 CRS-R's. Now we have explained this information in the manuscript. (Line 512-514)

Line 546: I am not entirely clear whether 20% is the threshold used to assign a node the role of a hub.

In the revised version we repeated the structural network analysis in the last section and replaced it with a weighted analysis. Therefore, SCs are no longer binarised. Indeed, we admit that it was rather incoherent to employ the weighted SC matrices to constrain the whole-brain network models along with the manuscript and then to binarise for this short network analysis. Note that this change supposes that the Supplemental Figures S10, S11 and S12 from the previous manuscript have been removed.

Line 567: Please clarify to whom a request should be addressed to if one wants to work with the data presented here.

We have included the e-mail address of the people to contact in case of requesting the data. Furthermore, part of the preprocessed data will be available in EBRAINS – the platform being developed by the Human Brain Project – in the near future.

Any translational value in these findings?

This study's main novelty comes from the investigation of the mechanisms underlying brain dynamics in different states of consciousness. This investigation can be only done by computational models which offer the possibility to study the local, global and network dynamics. So, the main translational value of the findings in understanding the mechanisms, dynamics, heterogeneity and network stability in pathological states of consciousness.

Even if the contribution of the model does not have an immediate application, there is a promising growing interest suggesting that whole-brain models might be a great help to support diagnosis and therapeutic interventions in disease (Deco et al. 2014, Gilson M et al. 2019, Kringelbach M et al., 2020). Indeed, as we indicated in the manuscript, enhancement of neural excitability in specific regions through therapeutic procedures, such as stimulation, may improve the conscious recovery process (Thibaut et al., 2019). Whole-brain models have been already proposed to investigate how a system reacts to a perturbation (Deco et al. 2018, Deco 2019). Our study investigated the altered dynamics in low-level states of consciousness and its properties, which offers insight into the dynamical mechanism of consciousness and the regions that support the network's stability. This information could be the crucial set of nodes for stimulation and recovery consciousness.

We believe that the reorganization of the discussion has clarified our results' translational value and novelty for understanding the study's contribution to the literature.

Supplementary materials:

Not entirely clear whether the healthy controls used for comparisons to DoC patients were scanned with the same MRI parameters as the DoC patients.

We have specified that the parameters of the MRI acquisition of the patients and controls were the same. (Supplemental line 39)

DoC patients have usually large lesions. How did the authors deal with this issue in terms of pre-processing the data spatially and temporally? If for example an ROI falls entirely within a lesion, what signal did the authors use in their analysis. Also, lesions can result in distortions following non-linear spatial normalization. How did the authors deal with this? Please provide some examples in supplementary materials.

This is a very relevant question and we agree that this point has to be addressed carefully, so the results are not affected by the lesions-noise artefacts present in the signals. As the reviewer indicated, DOC patients' lesions are usually large and sparse in the brain, so their location is difficult to determine. For this reason, pre-processing is crucial in this type of data. In our study, we considered the brain's lesions by applying a specific pre-processing step that aims to remove the lesion-driven artefacts by examining the independent component (ICs) obtained from the melodic FSL pipeline.

The pre-processing applied to the data consists of the standard pipeline, which contains motion-correction, non-brain removal, spatial smoothing, rigid-body registration and high-pass filter. At this point, we also included single-session ICA with automatic dimensionality estimation. This step aims to extract the ICs of each signal and, then, apply the FIX (FMRIB's ICA-based X-noiseifier), which allows us to remove the noise components and also the lesion-driven artefact noise. This procedure has been previously proposed by (Griffanti et al., 2014), showing that it can successfully remove the noise and clean the data. Indeed, this procedure has been applied in stroke patients, where after applying the FIX noisifier, the signals were able to reproduce the RSNs in an accurate manner (Carone D. et al., 2017). Notably, in our study, we performed this analysis for each individual subject, to check that the noise coming from the lesions was removed one by one. Furthermore, this procedure was performed by at least two people (ALG, RP and AE). Each component was classified by looking at the spatial map, the time series and the temporal power spectrum based on the criteria proposed in (Griffanti et al., 2014). Following the rationale underlying this method, the removal of lesions was not performed by ignoring the lesioned areas but removing the lesion-driven components.

In relation to the distortions that might be originated from the transformation, we followed the steps proposed in the papers by Andersson JLR et al. (2010) and (Griffanti et al., 2014) where they claimed that sometimes the linear transform is not sufficient to achieve good registration due to the differences between subjects. The local deformations permitted by a non-linear method may then do a better job. Some examples can be found in the following [link](https://fsl.fmrib.ox.ac.uk/fsl/fslwiki/FNIRT) (<https://fsl.fmrib.ox.ac.uk/fsl/fslwiki/FNIRT>).

Following the reviewer's proposal, we have included a new figure in the supplemental material (Fig. S10). We show two examples of two UWS patients with an evident lesion in the right hemisphere (A and C) and the spatial map of the component that reflects the lesion-driven artefact of that lesion (B and D), its time series and the spectral properties. Following the criteria established in (Griffanti et al., 2014) and the lesion's activation, those components were considered noise and removed from the time series using the FIX noisifier.

Reviewer #4 (Remarks to the Author):

Reviewer comments: "Loss of consciousness reduces the stability of brain hubs and the heterogeneity of brain dynamics" by López-González et al.

Summary

This manuscript describes a comparison between fMRI brain dynamics in healthy, wakeful participants and patients in low-level states of unconsciousness, such as disorders of consciousness or under anesthesia. The authors show that both pathological and pharmacological low-level states of consciousness presented altered network interactions and more homogenous and anatomically-constrained local dynamics than conscious states.

Overall impression

This study appears to support existing findings on network interactions associated with loss of consciousness. The tripartite conclusion of altered network interactions with more homogenous and structurally-constrained local dynamics, and less stability has validity through impressive statistical

analysis. However, the study's novelty must be clarified, as well as a stronger claim for the impact on consciousness as a cause of or a consequence of the observations.

We thank the Reviewer for his/her comments.

Specific comments with recommendations

In general, the paper is well thought-out with specific methodological contributions to the study of consciousness. The paper should be read for clarity, especially for run-on sentences and use of scientific jargon that does not always convey meaning (line 45, line 82, and throughout the discussion, etc.).

We have thoroughly revised the manuscript following several comments from referees. We very much appreciate all these recommendations and we hope the current version is substantially more clear and readable.

Consider rephrasing and tightening common phrases, “we were interested ...” with stronger language (i.e., “we investigated ... “ or “we analyzed...”).

Thank you for this comment. We have followed the reviewer's recommendations and we have modified some of the words such as 'interested' by 'investigated', and used active speech. Additionally, we have thoroughly revised the manuscript to improve readability and English grammar usage.

Abstract

1. What is the difference between structural and dynamical in the abstract (and elsewhere)?

To clarify the difference between the structural and dynamical analysis, we have included in the abstract the structural term and indicate that the structural analysis refers to the anatomy.

2. Similarly, please clarify about what you mean by “altered dynamics” especially in relation to changes in “local dynamics.” different regimes?

We agree with the reviewer that the term 'altered dynamics' is not specific and we have explained the changes in the local dynamics by explaining the meaning of each dynamical regime (Line 152-153).

3. I suggest adding the number of participants into your abstract.

We believe that adding the number of participants in the abstract is not needed considering that in this study we have used two datasets with different numbers of participants in each group. Also, the tight limit of 150 words imposed by the journal for the abstract already is quite a challenge. However, this information is indicated at the beginning of the results and in the methods part.

Introduction:

4. The introduction would be a good place to highlight what makes this study a novel contribution to the field. There has been a lot of work on coma, consciousness, and how they relate to anesthesia, and your study's innovations were not conveyed clearly here.

Thank you for this thoughtful comment. The main novelty in this study has been applying whole-brain computational models to understand the alterations in the functional mechanism of each brain state. This study goes beyond the literature in the topic of DOC, consciousness and anaesthesia by not only studying the properties of the brain signals – such as the correlations, brain states and their dynamics

– but also employing whole-brain models to investigate how structural, dynamical, local and network brain properties interplay in the different levels of consciousness. This work's main contribution in understanding the global and local dynamical alterations in the different states of consciousness. Especially, heterogeneity in brain dynamics is crucial to maintain consciousness and stress the hubs' role in supporting the stability of the network. Disentangle local and network effects and dynamical and structural ones can be only done through whole-brain computational models. We believe that this is a novel contribution to the field of neuroscience and the science of consciousness.

In the revised version, we have now clarified the importance of using whole-brain computational models to study how structural, dynamical, local and network brain properties interplay in the different levels of consciousness. Especially, the Discussion section has been very much reorganised to properly highlight the novelties of the present work and to better contextualise the results here reported within the field.

5. It is also advisable to clarify terms such as “low-level consciousness” and “awareness” or “arousal.” The distinction between arousal/level and content may be helpful here as described in anesthesia-induced loss of consciousness (see PMID 29937348; 28676745). How do those terms relate to your categories of consciousness?

We agree that the definition of the states and the differentiation of the states of consciousness requires clarification. We commented about the two dimensions that are commonly used to distinguish among levels of consciousness, wakefulness (i.e. arousal or level) and awareness (i.e. the content of consciousness). In particular, the levels of consciousness that we have studied are well classified along these two dimensions. For the sedation state, subjects showed no arousal nor awareness. In the case of patients with DOC – who unlike sedated subjects do show periods of wakefulness – mostly awareness is reduced. MCS patients show fluctuating signs of awareness, expressed as non-reflexive behavioural interactions with the world such as visual pursuit or response to the command, still unable to communicate. Although showing periods of arousal, UWS patients never exhibit voluntary behaviours indicative of conscious awareness, much like comatose patients. Following the reviewer's recommendation, we have included a short description of the dimensions (Line 42-43) and a short description of the low-level states of consciousness used during the study (Line 43-45).

6. “Deep sedation,” should be defined, even though the ASA provides a clinical definition. In the discussion, I would recommend the authors to rephrase the term “deep anesthesia.” Is this deep sedation, or general anesthesia? “Deep anesthesia” is too vague and conflates analgesia, dose of hypnotic movement, hemodynamic changes, amnesia, and consciousness. The fields of mechanisms of anesthesia is doing away with the concept of “deep” (PMID: 22011712).

As the paper that the reviewer suggests (Sleigh J. 2011), the depth of anaesthesia is a controversial concept that is still under debate. In the current study, we use data of subjects under deep sedation, which is a Ramsey score of 2 or 3 and not general anaesthesia. We agree with the reviewer that interchanging these definitions causes confusion. As our aim in this paper is not to define different levels of sedation, i.e. here we are not comparing multiple levels of sedation. Indeed, the recovered state is not considered sedation but a conscious state where the consciousness is not entirely recovered. So considering the fact that in this dataset we only considered the state of sedation, we have decided to refer to ‘sedation by propofol’ throughout the paper consistently.

Results

7. Again, be sure to defined "conscious wakefulness".

We have indicated in the introduction that conscious wakefulness refers to resting-state activity in the healthy participants (Line 77).

8. Also consider defining "synchronization dynamics" or "phase integration" in the Introduction.

As the reviewer indicated, the definition of the synchronization dynamics was mainly explained in the results part and, with more details, in the methods. As the paper is quite methodological and we wanted the introduction is understandable for a broad audience, we did not include the full explanation of the 'synchronization dynamics' in the introduction. But we agree that we should give a brief definition, so all readers can understand the line of the paper. With this aim, we defined the phase synchronization dynamics as the relationships between the phases of BOLD signals and, to motivate the use of the phases, the high resolution of this analysis has been indicated (Lines 79-82).

Discussion

9. Due to differences in anesthetics and their effects on level of sedation, references to "anesthesia" in this manuscript should be changed to "propofol anesthesia." A mention of the effect of different anesthetics would be an important addition to the limitations section (see, i.e., PMID 18988836).

Following your suggestion, we changed the term *anesthesia* by '*propofol anesthesia*'. Furthermore, as suggested, we include the limitation of the different anesthetics (Line 453-455)

10. There are many references to this study as "consistent" and "in line with" and "supports" previous studies. It is important to emphasize its novelty as well.

We agree with the reviewer that the contribution of our work was not clearly explained and emphasized in the manuscript. In the revised manuscript, we have re-written the discussion in order to make clear the contribution of our work and emphasize the novelty of the results and methods. We hope that the contribution is clearer in this new version.

11. Many brain regions are involved in loss of consciousness, as you mention. If anesthetics primarily affect the cortex or the thalamus, or brainstem disfunction, could the results you are seeing be a consequence or a cause of the loss of consciousness? How does the literature of thalamo-cortical or brainstem dysfunction (at the pontine level. See papers by Devor, Franks, Nir and others) relate to your findings?

Although the brainstem's role for consciousness is very relevant and still poorly defined, this was out of the scope of the study. For practical reasons, as the brainstem is not part of the chosen atlas, we did not look at brainstem properties. When comparing conscious wakefulness and sedation, our results show differences in the local bifurcation parameters in the cortex, particularly in the cingulum, insula, and some regions of the frontal part, paracentral and precentral. We also found important reductions in the local bifurcation parameters in the thalamus during propofol sedation, that was even bigger than the ones found in the cortex. Interestingly, the thalamus was not affected when comparing conscious wakefulness and recovery from propofol sedation. Our interpretation of these results is that these regions, and especially the thalamus in the thalamocortical loop, are responsible for

down-regulating the cortical network's activity. According to our results, this loop is impaired during sedation due to the alterations in the global and local dynamical mechanisms. This finding is consistent with previous work in anesthetics (see review Mashour et al., 2006) and DOC patients alike (Schiff et al., 2010). Although we can speculate that the thalamus is responsible for the loss of consciousness, due to the data and methods used in this study, we cannot test if the impairment of these areas is the consequence or cause of loss of consciousness. Hopefully, future work that includes experimental protocols suitable for answering this question will address it, firstly in animals and then, if a proper protocol is found, in humans.

We hope that this explanation is sufficiently answering your question. In case the reviewer had a more concrete concern, we would be happy to answer again.

12. The gender imbalance in several of your study groups (such as the anesthesia group) should be mentioned as a limitation.

It is true that the pharmacokinetics of propofol appears to be influenced by gender (Kodaka et al., 2005). Therefore, we have added this important point in the methods to notice this imbalance (Line 534). However, we would like to remark that the anesthetic dose was subject-specific, based on the behavioural assessment of the consciousness state, and therefore we trust that the brain dynamics are similarly affected for both genders and all subjects.

Methods

13. Rewrite line 363 for clarity.

Thank you. We agree with the reviewer that this sentence was not clear so we rewrote it in order to clarify the datasets presented (Line 499).

14. Did you analyze the DOC patients based on their underlying condition?

We did the analysis based on DOC patients' etiology classified into two groups, that is traumatic brain injury (i.e., TBI) and nonTBI (e.g., anoxia, hemorrhage, infection) patients. Due to limited sample sizes of the less frequent etiologies a more fine-grained distinction could not be made. We did not find any significant difference for the empirical measures and Hopf measures between the TBI and nonTBI patient groups. This indicates that, although the test might be underpowered at this stage with relatively low sample sizes, the altered brain network dynamics we observed are sensitive to consciousness levels and independent from the patient's underlying condition.

The statistics were computed by using a two-sample t -test.

Integration: $p=0.3049$

Segregation: $p=0.0160$

Phase-interaction fluctuations: $p=0.2674$

mean FCD: $p=0.2835$

Global coupling g : $p=0.9371$

15. *More details about the anesthetic procedure are necessary (apart from the reference in line 391 to footnote 60). For example, what time after anesthetic administration are recovery levels observed? Why are there so many female participants? What was the anesthetic dose?*

We have considered the reviewer's comment and decided to include a section in the supplementary material explaining the *sedation protocol* with the main details of the propofol administration, including the procedure to administer the propofol and determination of the sedation stage. During the recruitment for the anesthesia study, more females than males volunteered. The anesthetic dose was subject dependent, enough to reach the desired depth of anesthesia, but usually in the range of 3microg/ml for sedation.

In fact, it is not easy to find subjects who agree to undergo (around 1 to 2 hours) of propofol sedation. As it is well known that brain dynamics are altered highly during propofol anesthesia, we assumed the gender inequality does not significantly affect the study. We specified this factor in the paper's methods section.

16. *It would have been preferable to do a concurrent EEG to find correlates with fMRI findings and to determine more accurately the level of sedation.*

The data acquisition protocol for the anesthesia data is now described in the supplementary material. In this dataset, the level of sedation was assessed using the Ramsey scale, the standard assessment tool to identify sedation depth. Potentially, the assessment of sedation levels could have been more accurate when supplemented with EEG markers, although there is no consensus on the best approach to do so (Kreuzer et al., 2017). Even though, we will take into account the reviewer's suggestion and consider implementing multimodal EEG and fMRI acquisitions in the future to reduce potential uncertainty in the level of sedation in the following studies.

Updated figures:

Figure 1: We have changed the images adding more transparency to the boxplot, so each subject's point is properly visible now.

Fig. 1: Changes in global properties of phase-dynamics induced by loss of consciousness. **a** BOLD band-pass signals (0.04-0.07 Hz) for two sample ROIs. The instantaneous phases, $\phi_i(t)$ and $\phi_k(t)$, of each signal were computed using the Hilbert transform. **b** At each time frame, the interaction between ROIs was given by the instantaneous phase difference, $\Delta\phi_{jk}(t) = |\phi_j(t) - \phi_k(t)|$, which can be represented as vectors in the unit circle of the complex plane. **c** Phase-interaction matrices $P_{jk}(t)$ were calculated as the cosine of the phase difference, $\cos(\Delta\phi_{jk}(t))$, at time t . All global measures used afterwards were based on the phase-interaction matrices. **d-e** The structure of phase interactions was described in terms of the integration and the segregation of the time-averaged phase interaction matrix (see Methods). **f** We quantified the temporal fluctuations of the mean phase synchrony (i.e., the average over ROIs of matrix $P(t)$) through its temporal standard deviation. **g** To detect the existence of recurrent synchronization patterns, we computed the FCD comparing phase-interaction matrices at different times (see Methods). Briefly, the FCD represents the (cosine) similarities between phase-interaction matrices at times t and t^0 for all possible pairs (t, t^0) . The panel shows the average similarity for each experimental condition. In panels d-g, each dot represents a participant and the boxes represent the measure's distribution. Boxplots represent the mean of the measures' values with a 95% confidence interval (dark) and 1 SD (light). Differences between groups were assessed using one-way ANOVA followed by FDR p-value correction. *: $p < 0.05$; **: $p < 0.01$; ***: $p < 0.001$ (see Table 1 for details).

Figure 2: We have changed the images adding more transparency to the boxplot, so each subject's point is properly visible now.

Fig. 2: Fitting of global coupling parameter in the whole-brain network model. **a** The global coupling model parameter g scales the weights of the SC matrix. Low and high values of g represent networks where the functional connectivity is weakly and strongly coupled to the structural networks, respectively. **b** To estimate this global parameter, we sought the model that best reproduced the distribution of FCD values (fixing all other model parameters). **c** KS-distance between the empirical and the model FCD distributions, as a function of g , for one participant of each subject group (top: healthy controls and DOC patients; bottom: awake and anaesthetised subjects). Solid lines and shaded areas represent the mean and the standard error of the fitting curves over simulation trials. **d** Optimal global coupling g for all participants. In each panel, each dot represents a participant and the boxes represent the distribution of g . Boxplots represent the mean of the measures' values with a 95% confidence interval (dark) and 1 SD (light). Differences between groups were assessed using one-way ANOVA followed by FDR p-value correction. *: $p < 0.05$; **: $p < 0.01$; ***: $p < 0.001$. In panels c and d, we used the healthy structural connectivity as the underlying connectivity of all models.

Figure 3: we have added the panel g) that previously was included in the supplementary material (i.e. Figure S6 in the previous manuscript).

Fig. 3: Local bifurcation parameters of the whole-brain model. **a-d** Estimated bifurcation model parameters a for each of the 214 nodes (sorted by node strength). Bars indicate the mean \pm standard deviations across simulation trials. Results for low-level states of consciousness (MCS and UWS) are compared against the healthy controls in a and b. Results for anesthesia and recovery (S and R states) are compared to the initial awake state (W) in c and d respectively. **e** Ranked absolute parameter difference, Δa , for all the comparisons. **f** Spatial distribution of $\Delta a > 0.15$ in the brain for each of the group comparisons. **g** Relationship between the absolute difference Δa and the strength of each node. The absolute difference of the parameter a values between different groups in function of the strength of the nodes extracted from the SC of the healthy controls. ρ corresponds to the Pearson correlation.

Figure 4: This figure has been moved from the supplemental to the main text.

Fig. 4: **Eigendecomposition of the Jacobi matrix.** The eigenvectors of the Jacobi matrix (N -dimensional vectors) were sorted according to the real part of the associated eigenvalues (top insets), $\text{Real}(\lambda)$, and the strength of the nodes (right insets).

Figure 5: This figure has been updated considering the weighted analysis of the structural connectivities.

Fig. 6: **Disruption of the structural connectivity (SC) in DOC patients.** **a** SC matrices were averaged over subjects for each clinical group (CNT, MCS, and UWS). **b** Average node strength of each node for each group. Shaded areas represent the standard error across subjects. The grey dashed line corresponds to the threshold, $S = 4.5$, which determines the hub regions. **c** ROIs with a significant decrease of strength in patients as compared to controls (Wilcoxon rank sum test, followed by FDR correction). Left (green): CNT-MCS comparison; Right (red): CNT-UWS comparison. **d** Distribution of the node strength in the population average SCs. The distribution in controls displays a longer tail corresponding to hub regions, depicted in **e**. **f** k -density curves – average weight of links between regions with strength $S > S^0$ – show the loss of a rich-club structure in MCS and UWS patients. **g** Average link weight between hubs (i.e. regions with $S > 4.5$) in yellow and the average link weight between non-hubs regions (i.e. $S < 4.5$) in violet. **h** Distribution of the estimated bifurcation parameters a_i using the average SC for each clinical group (healthy, MCS, and UWS). **i** The variance of the distribution of parameters a_i for each clinical group. **j** Median of the absolute residuals of the linear relationship between the a_{eff} vs strength. ***: $p < 0.001$, Wilcoxon rank-sum test, followed by FDR correction.

REVIEWERS' COMMENTS:

Reviewer #1 (Remarks to the Author):

The authors have answered point by point all my comments and have explained the changes made in the manuscript in reaction to all my suggestions. The authors have corrected several errors in the notation and discuss some of the issues proposed by the reviewers. I just commented on two questions.

Q: There exist alternative definitions of Integration and Segregation. This fact should be emphasized and the choice better motivated.

--Indeed there exist alternative definitions of Integration and Segregation of functional connectivity matrices. These are based on weighted versions of transitivity (see Newman ME. The structure and function of complex networks. SIAM review. 2003;45(2):167–256.) and (global efficiency (Latora V, Marchiori M. Efficient behavior of small-world networks. Physical review letters. 2001;87(19):198701. pmid:11690461) to measure segregation and integration, respectively and have been used in computational models of the brain (for more complete implementation and discussion see Coronel-Oliveros C, Cofré R, Orío P (2021) Cholinergic neuromodulation of inhibitory interneurons facilitates functional integration in whole-brain models. PLoS Comput Biol 17(2): e1008737. <https://doi.org/10.1371/journal.pcbi.1008737>).

Q: I can understand that some sensitive data cannot be shared. However, in my opinion, the code used to obtain the results should always be available, readable, and usable without even need to request it.

--It is very important that you will make the code publicly available through GitHub and part of the preprocessed dataset available in EBRAINS in the near future.

Reviewer #3 (Remarks to the Author):

The authors have made a tremendous effort to improve the manuscript and as a result the manuscript is more readable and the findings more relatable. Moving some of the results from supplementary materials to the main manuscript improved the narrative. The authors have addressed all my concerns in a comprehensive manner so I recommend the article for publication.

Reviewer #4 (Remarks to the Author):

The manuscript has been improved substantially from the previous submission. I have no further comments.